# Expression of ENL YEATS domain tumor mutations in nephrogenic or stromal lineage impairs kidney development

Zhaoyu Xue[1,5], Hongwen Xuan[1,5], Kin Lau [2], Yangzhou Su[1], Marc Wegener[3], Kuai Li[1], Lisa Turner [4], Marie Adams [3], Xiaobing Shi [1] & Hong Wen [1]✉

Recurrent gain-of-function mutations in the histone reader protein ENL have been identified in Wilms tumor, the most prevalent pediatric kidney cancer. However, their pathological significance in kidney development and tumorigenesis in vivo remains elusive. Here, we generate mouse models mimicking ENL tumor (ENL[T]) mutations and show that heterozygous mutant expression in *Six2*[+] nephrogenic or *Foxd1*[+] stromal lineages leads to severe, lineage-specific kidney defects, both resulting in neonatal lethality. Six2-ENL[T] mutant kidneys display compromised cap mesenchyme, scant nephron tubules, and cystic glomeruli, indicative of premature progenitor commitment and blocked differentiation. Bulk and spatial transcriptomic analyses reveal aberrant activation of *Hox* and Wnt signaling genes in mutant nephrogenic cells. In contrast, Foxd1-ENL[T] mutant kidneys exhibit expansion in renal capsule and cap mesenchyme, with dysregulated stromal gene expression affecting stroma-epithelium crosstalk. Our findings uncover distinct pathways through which ENL mutations disrupt nephrogenesis, providing a foundation for further investigations into their role in tumorigenesis.

Organogenesis in the kidney is a complex process involving cellular interactions, intricate signaling pathways, and precise genetic and epigenetic programming. Disruptions to any of these processes can lead to developmental defects and diseases, such as cancer. The developing kidney in mammals comprises three main cell lineages: the ureteric epithelium, nephrogenic mesenchyme, and stromal cells. The metanephric kidney originates from the intermediate mesoderm (IM) through reciprocal inductive interactions between the ureteric bud (UB) and metanephric mesenchyme (MM), leading to the invasion of the UB into the mesenchyme and the proliferation and expansion of the primitive cap mesenchyme (CM)[1–3]. The CM cells possess self-renewal capacity, representing multipotent nephron progenitors (NPs)[4–6]. During kidney development, NP cells commit and undergo a mesenchymal-to-epithelial transition, giving rise to pretubular aggregates (PTA) and subsequently the renal vesicle (RV), marking the earliest nephron stage[7,8]. Progressing through segmentation and elongation, the RV eventually differentiates into various specialized epithelial cells, forming mature nephrons, including podocytes in the glomerulus, proximal tubules (PT), loop of Henle (LOH), and distal tubules (DT), while the UB becomes the collecting duct. Sustaining a delicate balance between cap mesenchyme cell differentiation and proliferation within the outer nephrogenic zone persists until the second postnatal day in mice[9,10], ensuring the development of a full complement of nephrons and a functional organ system. Furthermore, the differentiation and integration of stromal cells also play a crucial role in kidney development[11]. Besides the contribution to interstitial fibroblasts, renal capsule, pericyte, and mesangial cells in adult kidneys, stromal progenitors also produce signals during development that regulate nephron progenitor cell renewal, ureteric branching, and kidney morphogenesis[12–16].

[1]Department of Epigenetics, Van Andel Institute, Grand Rapids, MI 49503, USA. [2]Bioinformatics and Biostatistics Core, Van Andel Institute, Grand Rapids, MI 49503, USA. [3]Genomics Core, Van Andel Institute, Grand Rapids, MI 49503, USA. [4]Pathology Core, Van Andel Institute, Grand Rapids, MI 49503, USA. [5]These authors contributed equally: Zhaoyu Xue, Hongwen Xuan. ✉e-mail: hong.wen@vai.org

Wilms tumor is the most common pediatric kidney cancer affecting one in 10,000 children in North America. It is a prototypical embryonal malignancy closely resembling the histological features of the developing kidney[7]. Wilms tumor arises from the failure of embryonic nephrogenic cells to undergo terminal differentiation, resulting in persistent regions of embryonic tissue containing blastemal cells exhibiting varying degrees of differentiation[7,17,18]. Until recently, our understanding of genetic alterations associated with Wilms tumor was primarily confined to genes known to be involved in kidney development, such as *WT1*, as well as Wnt-activating mutations in *CTNNB1* and *AMER1*[17]. However, comprehensive genomic analyses have recently unveiled novel mutations in high-risk Wilms tumor cases, indicating a high degree of genetic heterogeneity in this malignancy[19–25]. Notably, many of these mutated genes converge on pathways involved in the epigenetic regulation of transcription, including *ENL*, *CDC73*, *CTR9*, *BCOR*, *BCORL1*, *EP300*, and *CREBBP*, underscoring the critical role of proper transcriptional regulation in maintaining normal kidney development. Further exploration of the mechanisms underlying how mutations in epigenetic regulators disrupt kidney development holds great promise for advancing our understanding of Wilms tumor etiology and facilitating the development of novel therapeutic strategies.

ENL (also known as MLLT1 or YEATS1) functions as a transcriptional coactivator primarily associated with the super elongation complex (SEC)[26–31] and the histone H3K79 methyltransferase DOT1L complex[32–34], regulating transcription elongation. It contains an evolutionary conserved YEATS (Yaf9, ENL, AF9, Taf14, Sas5) domain, which we previously uncovered as a reader module for histone acylation[35–40]. Recent studies have identified hotspot somatic mutations of *ENL* in 4–9% of Wilms tumor[19,20]. These mutations, hereafter referred to as *ENL* tumor mutations or *ENL^T* mutations, are unique small in-frame insertions or deletions in one allele of *ENL*, altering the coding sequence of an eight-amino-acid polypeptide (aa111-118) within the C-terminus of the YEATS domain[19,20]. A prominent gene-expression signature in Wilms tumors with *ENL^T* mutations is the aberrant activation of a subset of the *HOX* genes[19,20], which encode transcription factors crucial for embryonic patterning. Notably, previous genetic studies have shown that the posterior *Hox* genes are essential for proper lineage selection and maintenance during mouse kidney development[14,41,42].

Previous studies from us and others have unveiled that ENL^T mutations are gain-of-function mutations that promote chromatin recruitment of ENL and its associated transcription machinery, leading to aberrant activation of genes important in cell fate determination[43]. Moreover, expression of ENL^T mutants in mouse embryonic stem cells results in dysfunctional nephrogenesis and the formation of Wilms tumor-like blastema structures in kidney organoid[43]. While these cell culture studies have begun to elucidate the oncogenic potential of ENL^T mutations, the pathogenic role of these mutations in kidney development and tumorigenesis in vivo are less explored.

In this study, we have generated conditional knock-in mouse models mimicking two patient-derived *ENL^T* mutations. We show that heterozygous expression of ENL^T mutants in either nephrogenic or stromal lineage profoundly impairs nephrogenesis, resulting in neonatal lethality. Through comprehensive characterization, we delineate distinct phenotypes of mice expressing ENL^T mutants in nephrogenic and stromal lineages. Bulk and spatial transcriptomic analyses identify gene expression alterations underlying the deficits in kidney development. Collectively, our study sheds light on how these ENL YEATS mutations impede nephrogenesis through different pathways in nephrogenic and stromal compartments, paving the way for further investigation into their roles in tumorigenesis.

## Results

### Conditional knock-in mouse models for ENL YEATS mutations
To investigate the pathogenic role of the ENL YEATS domain tumor mutations in kidney development, we generated conditional knock-in mouse models for two frequent mutations: *Enl^I117_118insNHL* (hereafter named *Enl^T1*), the most prevalent *ENL* insertion mutation found in Wilms tumor, and *Enl^I111_113NPP>K* (also named as *Enl^T3*), a small deletion mutation[19,20] (Fig. 1a). The Cre/loxP-based *Enl^T* mutant mouse models (*Enl^T-fl*) were generated in the C57BL/6 J background using CRISPR/Cas9 gene targeting technology (Supplementary Fig. 1a).

In these mouse models, a wildtype *Enl* allele was replaced with an engineered allele that consists of a loxP-flanked cassette containing a wildtype *Enl* exon 4, the coding sequence of *Enl* exons 5–12, and a BGH polyadenylation signal (BGHpA), followed by a mutant *Enl* exon 4 encoding a mutant ENL (ENL^T1 or ENL^T3). Without Cre recombinase, wildtype *Enl* is transcribed from the floxed sequence. In the presence of Cre-recombinase activity, recombination between the two loxP sites deletes the wildtype *Enl* coding sequence, resulting in the expression of mutant *Enl* at a physiological level driven by endogenous *Enl* promoter (Fig. 1b). Short- and long-range genomic PCRs (Supplementary Fig. 1b, c) and DNA Sanger-sequencing of the PCR products confirmed the successful knock-in of the designed targeting alleles.

To validate the Cre-mediated recombination and assess mutant expression, we isolated tail fibroblasts from wildtype (WT, *Enl^+/+*), *Enl^T-fl/+* (het), and *Enl^T-fl/T-fl* (homo) mice and infected the cells with CMV-Cre virus (Supplementary Fig. 2a). As antibodies detecting endogenous ENL in mice were not available, we assessed the expression of wildtype *Enl* and *Enl^T* mutants by semiquantitative RT-PCR and DNA sequencing in cells with different genotypes. The results demonstrated that our conditional knock-in mouse models worked as expected (Supplementary Fig. 2b–f). The *Enl^T1* and *Enl^T3* alleles were expressed at levels comparable to the wildtype *Enl* allele in CMV-Cre-transduced heterozygous *Enl^T-fl/+* fibroblasts (Supplementary Fig. 2c, d).

### ENL^T mutants disrupt renal development in whole embryo
To induce ubiquitous expression of mutant ENL in mice, we generated a small cohort of mice by crossing *Enl^T-fl/+* mice with *CMV-Cre^tg/+* mice[44]. However, we failed to obtain live ENL^T mutant pups (*CMV-Cre^tg/+*; *Enl^T-fl/+*) even with just one copy of *Enl^T1* or *Enl^T3* mutation induced (Fig. 1c). We collected developing kidneys from live mutant fetuses at E18.5 and found that ENL^T mutant kidneys were significantly smaller than those from Cre only, *Enl^T-fl* or completely wildtype littermate embryos (Fig. 1d).

Histologically, wildtype, Cre only, and *Enl^T-fl* embryonic kidneys showed normal morphology with various developing nephron structures, including branching ureteric buds (UB) surrounded by cap mesenchyme (CM) in the nephrogenic zone, differentiating nephron structures of the comma-shaped body (CSB) and S-shaped body (SSB), and more mature nephrons consisting of glomeruli and connected tubules (Fig. 1e and Supplementary Fig. 3a). In contrast, ENL^T1 and ENL^T3 mutant kidneys lacked mature nephron structures, such as glomeruli and proximal tubules (PT); instead, they exhibited cyst-like structures, likely derived from failed glomerular development (Fig. 1d, e and Supplementary Fig. 3b). These ENL^T mutant kidneys also exhibited disorganized nephrogenic zone, but with recognizable CSB and SSB structures.

WT1 is expressed in cap mesenchyme, renal vesicles, the proximal regions of CSB and SSB, and podocytes in glomerulus[45,46], as shown in the immunostaining of control kidneys (Fig. 1f and Supplementary Fig. 3c). Immunostaining of WT1 further validated disorganized nephrogenic zone and cystic dilation of the Bowman's space in the ENL^T mutant kidneys (Fig. 1g and Supplementary Fig. 3d). Most cysts were lined up by a single layer of WT1 positive cells, and some of them contained atrophic glomeruli.

Immunostaining of E-Cadherin (E-Cad) in the control kidneys showed strong staining of epithelial cells in ureteric buds, collecting ducts and distal tubules (DT) with light staining of proximal tubules (PT) (Fig. 1h and Supplementary Fig. 3e). However, although collecting ducts and some distal tubules were present, proximal tubules were almost completely lost in the mutant kidneys (Fig. 1i and

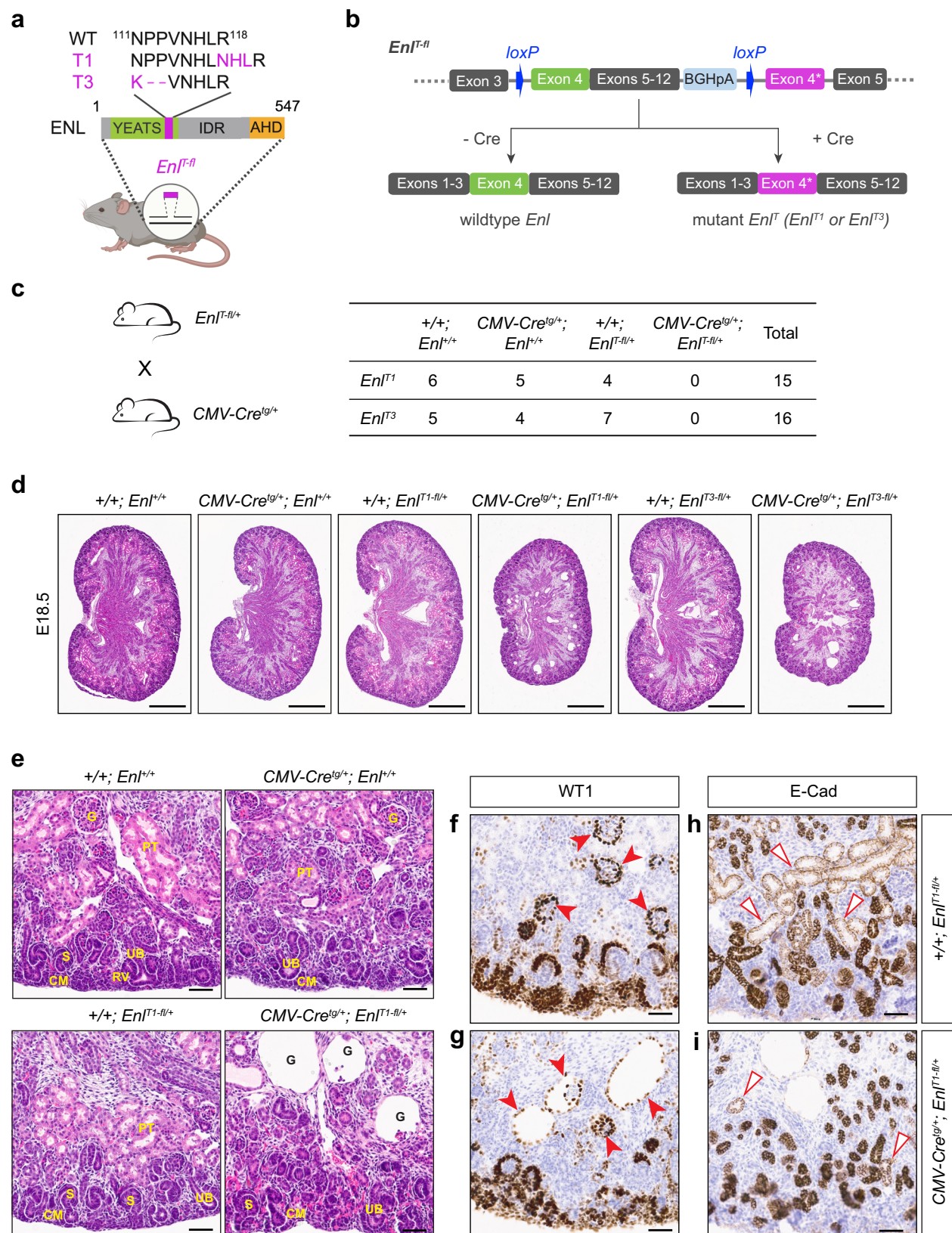

Together, these findings suggest that ENL YEATS domain tumor mutations have a deleterious impact on kidney development, and the analogous phenotypes between the two mutants indicate that the insertion mutant (ENL[T1]) and deletion mutant (ENL[T3]) likely function through a similar mechanism, consistent with our previous study in cultured cells[43].

## ENL[T] mutants in the Six2[+] nephrogenic lineage impede nephrogenesis

Given the embryonic lethality associated with whole-body expression of ENL[T] mutants, we chose to introduce Enl[T] mutations specifically to the nephrogenic lineage to evaluate their impact on nephrogenesis. The transcription factor SIX2 is predominantly expressed in the cap

**Fig. 1 | Generation of conditional knock-in (cKI) *Enl^{T-fl}* mouse models.**
**a** Schematic of ENL[T1] (T1) and ENL[T3] (T3) mutations in mouse models. IDR, intrinsic disordered region; AHD, ANC-1 homology domain. Created in BioRender. Wen, H. (2025) https://BioRender.com/w38y497. **b** Schematic of cKI allele of *Enl^{T1}* or *Enl^{T3}*. **c** Lethality induced by the expression of ENL^T mutants driven by *CMV-Cre^{tg}*. Schematic of the breeding strategy on the left. The table lists the numbers of viable progeny with all expected genotypes. **d** Hematoxylin and eosin (H&E) staining of kidney sections from E18.5 embryos with indicated genotypes. Scale bars, 0.5 mm. **e** Zoomed-in images of H&E-stained sections showing histological changes

observed in ENL[T1] mutant kidney at E18.5. CM cap mesenchyme, G glomerulus, PT proximal tubule, RV renal vesicle, S S-shaped body, UB ureteric bud.
**f, g** Immunostaining for WT1 of E18.5 kidney sections from *Enl^{T1-fl/+}* (**f**) and *CMV-Cre^{tg/+};* *Enl^{T1-fl/+}* (**g**) embryos. Solid red arrowheads indicate developing glomeruli at various stages. **h, i** Immunostaining for E-Cadherin (E-Cad) of E18.5 kidney sections from *Enl^{T1-fl/+}* (**h**) and *CMV-Cre^{tg/+}; Enl^{T1-fl/+}* (**i**) embryos. Red open triangles indicate proximal tubules with light E-Cad staining. Scale bars in **e–i**, 50 μm. For (**d–i**), similar results were obtained in three independent experiments.

mesenchyme cells adjacent to the inductive ureteric epithelium[47]. *Six2* expression initiates around E10.5 within the metanephric mesenchyme, and *Six2^+* cap mesenchymal cells represent a multipotent nephron progenitor population, which gives rise to all cell types in the main body of nephrons, including glomerular podocytes, proximal tubules, distal tubules, and the loop of Henle[4] (Fig. 2a). We employed the *Six2-TGC^g* BAC transgenic strain with a transgene expressing GFP-tagged Cre under the control of the *Six2* promoter integrated into chromosome 1, thereby leaving the expression of the endogenous *Six2* unaffected[4].

Upon crossing *Six2-TGC^g* mice with *Enl^{T-fl}* mice, we observed the birth of *Six2-TGC^{tg/+}; Enl^{T-fl/+}* mutant (hereafter referred to Six2-ENL^T) and ENL^{WT} pups with other genotypes (+/+; *Enl^{+/+}, Six2-TGC^{tg/+}; Enl^{+/+}*, and +/+; *Enl^{T-fl/+}*) at Mendelian ratios. However, neonatal mortality was observed in Six2-ENL^T mutant pups (Supplementary Fig. 4a). Both Six2-ENL[T1] and Six2-ENL[T3] mutant pups succumbed shortly after birth due to a complete inability to produce urine, resulting in significantly deflated bladders compared with other ENL^{WT} pups (Fig. 2b). We collected kidneys at E16.5 and E18.5 during embryonic development and at birth (P0) and found that the Six2-ENL^T mutant kidneys were markedly reduced in both weight and size compared to ENL^{WT} counterparts at all stages examined (Fig. 2b–e and Supplementary Fig. 4b–e).

The expression of wildtype and mutant *Enl* in the embryonic kidneys was validated by RT-PCR with allele-specific primers (Supplementary Fig. 4f, g). Recently, it was reported that the *Six2-TGC^g* BAC transgenic strain contains the *Six3* gene within the transgene and exhibits aberrant *Six3* expression driven by a *Six2* distal enhancer, which may cause reduced nephron endowment[48]. In our study, we did not observe obvious differences in embryonic kidney development between *Six2-TGC^{tg/+}; Enl^{+/+}* and +/+; *Enl^{+/+}* or +/+; *Enl^{T1-fl/+}* (Supplementary Fig. 4h, i). In subsequent studies, we used *Six2-TGC^{tg/+}; Enl^{+/+}* kidneys as ENL^{WT} controls.

At birth (P0), Six2-ENL^T mutants exhibited renal dysplasia, manifesting phenotypes similar to those observed in CMV-Cre-driven ENL^T mutant kidneys (Fig. 2e). The impairment in nephrogenesis was more pronounced in Six2-ENL[T1] mutants than in Six2-ENL[T3] mutants. Nevertheless, in both mutant kidneys, developed nephron tubules were scarce, and normal mature glomeruli were nearly absent (Fig. 2e), indicating a blockade in nephrogenesis at the maturation stage. Defects in nephrogenesis were developed early during embryonic development in Six2-ENL^T mutants (Supplementary Fig. 5a–l). Typically, by E16.5, the ureteric bud has undergone extensive branching, and nephrogenesis is well advanced. In ENL^{WT} kidneys, nephron structures at various stages–from cap mesenchyme to mature nephrons–were well-arranged perpendicular to the renal cortical surface in the cortex (Supplementary Fig. 5a–d). In contrast, Six2-ENL^T mutants exhibited fewer differentiating and differentiated nephron structures, and thinner cap mesenchyme with disrupted alignment (Supplementary Fig. 5e–l). Furthermore, these defects became more pronounced at E18.5 (Supplementary Fig. 5m–x).

Next, we performed immunofluorescence staining for markers of various nephron structures to further evaluate the developmental abnormalities in Six2-ENL^T mutant kidneys (Fig. 2f–v and Supplementary Figs. 6–8). SIX2 is predominantly expressed in the cap

mesenchyme cells[47]. We first stained SIX2 for the multipotent nephron progenitor population. The numbers of SIX2^+ cap mesenchyme were greatly reduced in Six2-ENL^T mutant kidneys (Fig. 2f, g, n and Supplementary Figs. 6a–f, 7a–d, and 8a). Additionally, the numbers of SIX2^+ nephrogenic progenitor cells in each cap mesenchyme were significantly reduced, leading to a thinner layer of SIX2^+ cap mesenchyme surrounding KRT8^+ ureteric buds compared to that in ENL^{WT} counterparts (Fig. 2o and Supplementary Fig. 8b). SIX2 plays a crucial role in maintaining a functional pool of self-renewing nephron progenitor population. Depletion of *Six2* leads to rapid loss of the progenitor pool and early termination of nephrogenesis[49]. Therefore, our results indicate that the expression of ENL^T mutants in *Six2^+* nephron progenitor cells compromises the initial steps in nephrogenesis.

We then assessed the formation of nascent nephrons in Six2-ENL^T mutants by immunostaining of the neural cell adhesion molecule (NCAM) and WT1. NCAM is a marker for nephron progenitors and nascent nephrons[50], while WT1 is expressed in nephron progenitors, the proximal segment of nascent nephrons, and podocytes[45,46]. Compared to their ENL^{WT} counterparts, the Six2-ENL^T mutant kidneys exhibited a more primitive state, characterized by a thicker nephrogenic zone (Fig. 2p and Supplementary Fig. 8c). In Six2-ENL^T mutants, cap mesenchyme exhibited low WT1 and NCAM staining signals (Fig. 2h, i and Supplementary Figs. 6g–l, 7e–h). The mutant kidneys displayed a reduced number of normal WT1^+ NCAM^+ nascent nephron structures, including RV and CSB/SSB (Fig. 2q, r and Supplementary Fig. 8d, e). Early podocytes in glomerular cleft were observed in Six2-ENL^T mutants, but they failed to develop into more mature glomeruli beyond these stages (Fig. 2u and Supplementary Figs. 6g–l, 7e–h, 8h).

Virtually all rudimental glomeruli in Six2-ENL^T mutant displayed cystic features with marked dilation of the Bowman's space, particularly evident in Six2-ENL[T1] mutants (Fig. 2e, v and Supplementary Fig. 8i). Co-staining of WT1 and PODXL, a podocyte marker[51], indicated that either early nephron did not produce enough podocytes in differentiating glomeruli or podocytes were depleted in the Six2-ENL^T mutants. The alignment of PODXL^+ and WT1^+ cells in Six2-ENL^T glomeruli was abnormal (Supplementary Fig. 8j, k). Additionally, we detected a reduced number of Ki-67^+ cells in Six2-ENL[T1] mutant kidneys, suggesting decreased cell proliferation (Supplementary Fig. 8l).

We also examined the development of elongating tubules by staining with Lotus tetragonolobus lectin (LTL), a marker for proximal tubules[52], and E-Cad, an epithelial marker[53]. LTL is predominantly located on the apical membrane of proximal tubules, while E-Cad is present on the basolateral membranes of all tubular segments, most abundantly in distal tubules and at very low levels in the proximal tubules[53]. Consistent with our observations in H&E staining, LTL staining indicated that renal tubules were well developed in ENL^{WT} kidneys, whereas Six2-ENL^T mutants exhibited very few LTL^+ proximal tubule-like structures (Fig. 2j, k, s and Supplementary Figs. 6m–r, 7i–l, 8f). Staining with SLC12A3 and E-Cad, markers of the apical and basolateral membranes of distal tubules, respectively, revealed compromised distal tubule development in Six2-ENL^T mutants (Fig. 2l, m, t and Supplementary Figs. 6s–x, 7m–p, 8g). In contrast, collecting duct

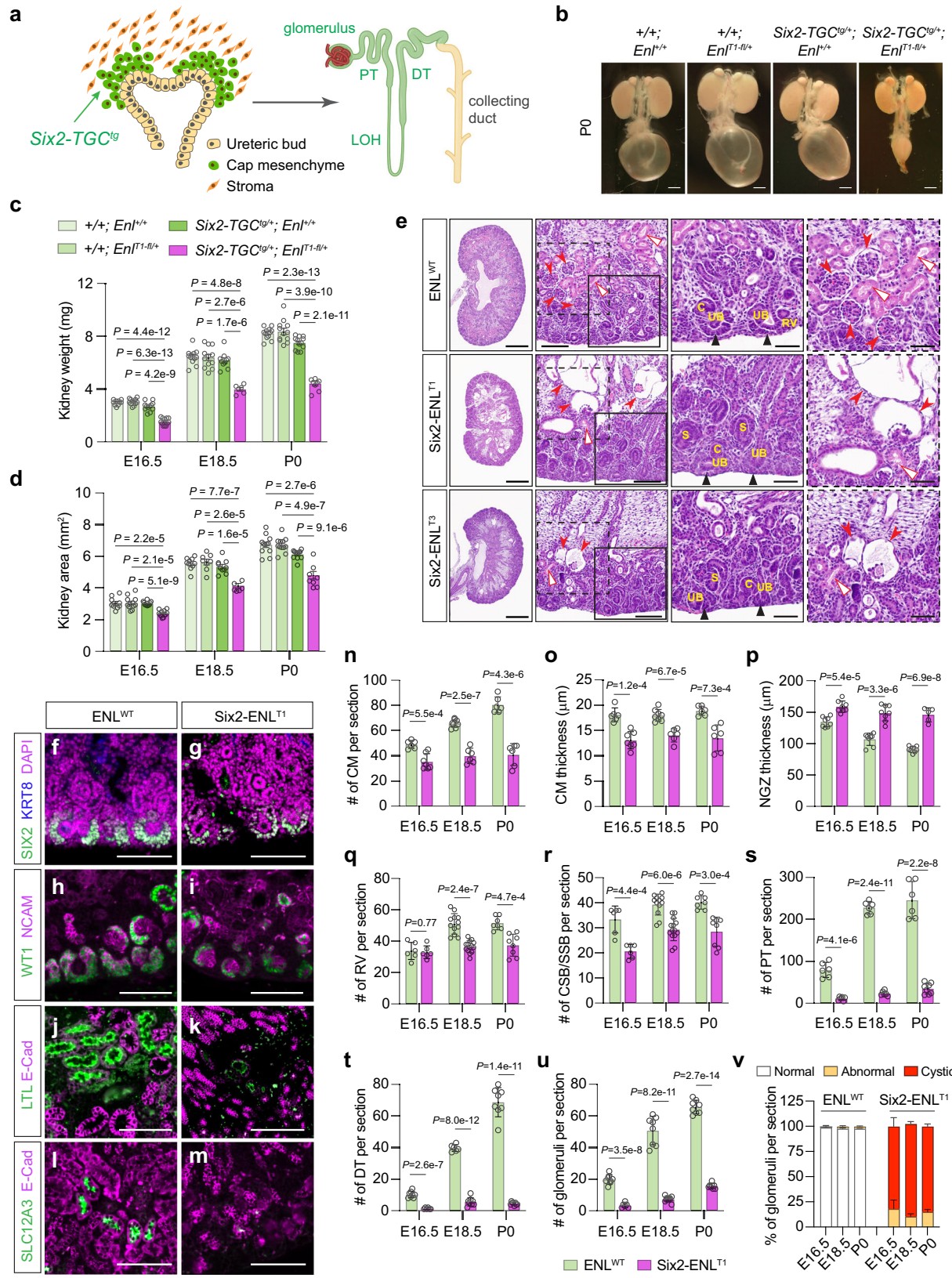

development was not significantly affected in Six2-ENL[T] mutants, as indicated by the relatively normal E-Cad staining (Fig. 2j–m and Supplementary Figs. 6m–x, 7i–p). Together, these H&E and immunostaining results suggest that the expression of ENL[T] mutants in the *Six2*[+] progenitor population impedes nephrogenesis at multiple stages.

## Global gene expression changes in Six2-ENL[T] mutant kidneys

To investigate the molecular mechanism by which ENL[T] mutations perturb kidney development, we first performed bulk RNA sequencing (RNA-seq) analysis in *Six2-TGC*[tg/+] (ENL[WT]) and Six2-ENL[T] mutant kidneys collected at E18.5. We identified 669 upregulated genes and 1031 downregulated genes in Six2-ENL[T1], and 987 upregulated genes and

**Fig. 2 | Induced ENL$^{TI}$ mutant expression in nephron progenitors impedes embryonic kidney development. a** Schematic of *Six2-TGC$^g$* expression in nephrogenesis. *Six2$^+$* cells and nephron derivatives are highlighted in green. DT distal tubule, LOH loop of Henle, PT proximal tubule. Created in BioRender. Wen, H. (2025) https://BioRender.com/g02w127. **b** Bright-field images of P0 kidneys with indicated genotypes. Scale bars, 1 mm. **c, d** Kidney weight (**c**) and size (**d**). Sample numbers at E16.5, E18.5 and P0: +/+; *Enl$^{+/+}$* (*n* = 10, 12, 12); +/+; *Enl$^{TI-fl/+}$* (*n* = 12, 12, 12); *Six2-TGC$^{g/+}$*; *Enl$^{+/+}$* (*n* = 12, 10, 12); and *Six2-TGC$^{g/+}$*; *Enl$^{TI-fl/+}$* (*n* = 12, 6, 8). **e** H&E staining of ENL$^{WT}$ (*Six2-TGC$^{g/+}$*; *Enl$^{+/+}$*), Six2-ENL$^{TI}$ and Six2-ENL$^{T3}$ P0 kidney sections. Solid line boxes are magnified, highlighting the nephrogenic zone (NGZ), and dash line boxes are magnified, focusing on glomerulus and PT. Red arrowheads indicate glomeruli; red open triangles indicate PT; and black triangles indicate cap mesenchyme (CM). C comma-shaped body, RV renal vesicle, S S-shaped body, UB ureteric bud. Scale bars: 0.5 mm (first column), 0.1 mm (second column), and 50 μm (last two zoom-in columns). **f–m** Immunofluorescence staining of SIX2 and KRT8 (**f, g**), WT1 and NCAM (**h, i**), LTL and E-Cad (**j, k**), and SLC12A3 and E-Cad (**l, m**) in P0 kidney sections. Scale bars, 100 μm. **n–u** Quantification of nephron structures per section. CM numbers (**n**) and thickness (**o**) (*n* = 6, 8, 6 ENL$^{WT}$ and 7, 6, 6 ENL$^{TI}$); NGZ thickness (**p**) (*n* = 8, 8, 8 ENL$^{WT}$ and 8, 8, 6 ENL$^{TI}$); numbers of RV (**q**) and CSB-SSB (**r**) (*n* = 6, 12, 7 ENL$^{WT}$ and 6, 14, 8 ENL$^{TI}$), PT (**s**) (*n* = 6, 6, 6 ENL$^{WT}$ and 6, 6, 8 ENL$^{TI}$), DT (**t**) (*n* = 8, 6, 8 ENL$^{WT}$ and 8, 6, 8 ENL$^{TI}$) and glomeruli (**u**) (*n* = 8, 8, 8 ENL$^{WT}$ and 6, 10, 8 ENL$^{TI}$). **v** Percentage of glomeruli with normal, abnormal and cystic morphologies in the same set of ENL$^{WT}$ and Six2-ENL$^{TI}$ kidneys as in (**u**). Data represent mean ± s.d.; two-tailed unpaired *t*-test. Source data are provided as a Source Data file.

1070 downregulated genes in Six2-ENL$^{T3}$ relative to ENL$^{WT}$ kidneys (fold change >1.5, FDR <0.05) (Supplementary Fig. 9a and Supplementary Data 1). Global gene expression changes in Six2-ENL$^{TI}$ and Six2-ENL$^{T3}$ over ENL$^{WT}$ kidneys exhibited a high correlation (PCC = 0.84, *P* value <2.2e-16), with 605 upregulated genes and 909 down-regulated genes overlapped between the two mutant kidneys (Supplementary Fig. 9b, c).

Gene set enrichment analyses (GSEA)[54] unveiled that gene expression changes caused by ENL$^T$ mutations in HEK293 cells[43] were well resembled in Six2-ENL$^T$ mutant kidneys (Supplementary Fig. 9d, e). Furthermore, we analyzed RNA-seq datasets of Wilms tumor samples from the TARGET study[19] and defined differentially expressed genes (DEGs) between tumors with and without *ENL* mutations (Supplementary Data 2). GSEA showed that the upregulated genes in ENL-mutant Wilms tumors were positively enriched, and downregulated genes in tumors were negatively enriched, in Six2-ENL$^T$ mutant mouse kidneys (Supplementary Fig. 9d, e), indicating potential clinical relevance of our mutant mouse models. Gene ontology (GO) analysis revealed that genes upregulated in Six2-ENL$^T$ mutant kidneys were enriched in biological process terms involved in anterior/posterior pattern specification, embryonic and organism development, immune response, and ion transport; whereas the downregulated genes were involved in various transport and metabolic processes, reflecting developmental defects and loss of kidney function at the organ level (Supplementary Fig. 9g, h and Supplementary Data 3).

Dysregulation of *HOX* genes represents a key gene-expression signature in human Wilms tumors bearing *ENL$^T$* mutations[20]. Nearly all *Hox* genes are expressed during kidney development[55]. Notably, posterior *Hox* genes, such as *Hox9* to *Hox11*, play pivotal roles in kidney development[42]. In our RNA-seq analyses, we observed upregulation of numerous *Hox* genes in both Six2-ENL$^{TI}$ and Six2-ENL$^{T3}$ mutant kidneys (Supplementary Fig. 9f, i, j and Supplementary Data 3), largely resembling what was observed in Wilms tumors with *ENL* mutations[19,20].

Among the downregulated genes in Six2-ENL$^T$ mutant kidneys, we observed a marked reduction in the expression of many marker genes for differentiating and mature nephron structures[56,57]. These include podocyte markers *Nphs1/2*, *Podxl*, *Ptpro*, and *Mafb*, proximal tubule markers *Aqp1*, *Hnf4a*, *Spp2*, *Kap*, and *Slc34a1*, and distal tubule & loop of Henle markers *Clcn5*, *Slc12a1*, *Slc12a3*, *Umod*, and *Cldn19* (Supplementary Fig. 9i–m). As the gene expression profiles were derived from the whole kidney, the reduced expression of these marker genes primarily signifies the absence of these cell types or structures in the mutant kidneys.

Compared with the primary downregulation of marker genes for mature nephron structures, the expression changes of nephron progenitor genes were rather complex. While downregulation of nephron progenitor "stemness" genes *Six2*, *Cited1*, *Crym*, *Osr1*, *Spock2*, and *Eya1* was observed in Six2-ENL$^T$ mutant kidneys, other genes expressed in committing progenitors within pretubular aggregates and renal vesicles, including *Wnt4*, *Sfrp2*, *Bmper*, and *Tmem100*, exhibited

upregulation[56,57] (Supplementary Fig. 9i, j, n). These genes play crucial roles in regulating the self-renewal and commitment of nephron progenitor cells during nephrogenesis. Therefore, the observed changes in Six2-ENL$^T$ mutant kidneys suggest compromised self-renewal and premature commitment of nephron progenitor cells, alongside blocked terminal differentiation.

## Gene expression in *Six2$^+$* progenitor-derived ENL-mutant cells

In Six2-ENL$^T$ kidneys, ENL$^T$ mutants were expressed only in the nephron progenitor cells and their progeny, but not in the ureteric epithelium and stromal lineage cells. To specifically investigate gene expression changes in ENL$^T$ mutant expressing cells, we utilized Ai14D[58], a Cre reporter strain containing a loxP-flanked STOP cassette upstream of the CAG promoter-driven tdTomato (*LSL*-tdT), to trace ENL$^T$ mutant cells in embryonic kidneys. By crossing *Six2-TGC$^{g/+}$* mice with homozygous *LSL-tdT$^{tg}$*; *Enl$^{T1-fl}$* mice, *Six2*-Cre induced the expression of tdT and ENL$^{TI}$, resulting in permanent fluorescence labeling in ENL-mutant nephron progenitors as well as their derived cells.

We utilized fluorescent-activated cell sorting (FACS) to isolate tdT$^+$ cells from Six2-ENL$^{TI}$ kidneys and conducted RNA-seq analysis of the sorted cells (Fig. 3a). The tdT$^+$ cells isolated from *Six2-TGC$^{g/+}$*; *LSL-tdT$^{tg/+}$* kidneys were used as ENL$^{WT}$ controls. We collected cells at both E14.5 and E18.5 to assess ENL$^{TI}$ mutation-induced gene expression changes at different developmental stages. Overall, the ratios of tdT$^+$ cells were comparable between ENL$^{WT}$ and ENL$^{TI}$ in E14.5 kidneys (31.7% in ENL$^{WT}$, 31.6% in ENL$^{TI}$), whereas this ratio was lower in ENL$^{TI}$ kidneys at E18.5 (26.9% in ENL$^{TI}$ *vs.* 33.2% in ENL$^{WT}$) (Supplementary Fig. 10a), indicating diminished cell populations of nephron structures in ENL$^{TI}$ mutant kidneys at the later stage.

In E14.5 kidney rudiments, we identified 935 upregulated genes and 497 downregulated genes in tdT$^+$ cells from ENL$^{TI}$ mice relative to the ENL$^{WT}$ control mice. The number of DEGs in ENL$^{TI}$ mutant cells from E18.5 kidneys was significantly higher, with 1273 and 1125 genes up- and downregulated, respectively (Fig. 3b–d and Supplementary Data 4). Similar to what we observed in the RNA-seq analysis of whole kidneys (Supplementary Fig. 9d, e), up and down DEGs in HEK293 cells expressing human ENL$^{TI}$ and in *ENL*-mutant Wilms tumors were enriched in the sorted ENL$^{TI}$ mutant nephron progenitors and their derived cells from E14.5 and/or E18.5 kidneys (Fig. 3e and Supplementary Fig. 10b). Additionally, genes associated with cell fate commitment were positively enriched in ENL$^{TI}$ mutant cells at both stages. Conversely, genes associated with kidney development and renal system process exhibited negative correlations with the gene expression changes detected in ENL$^{TI}$ mutant cells at E18.5, suggesting severe kidney developmental defects at this late embryonic stage (Supplementary Fig. 10b). The developmental anomalies may have triggered the expression of genes involved in the activation of immune responses in E18.5 ENL$^{TI}$ mutant nephron cells, a phenomenon not observed at E14.5 (Supplementary Fig. 10b). Moreover, the top GO terms enriched in the DEGs in sorted ENL$^{TI}$ mutant cells were similar to

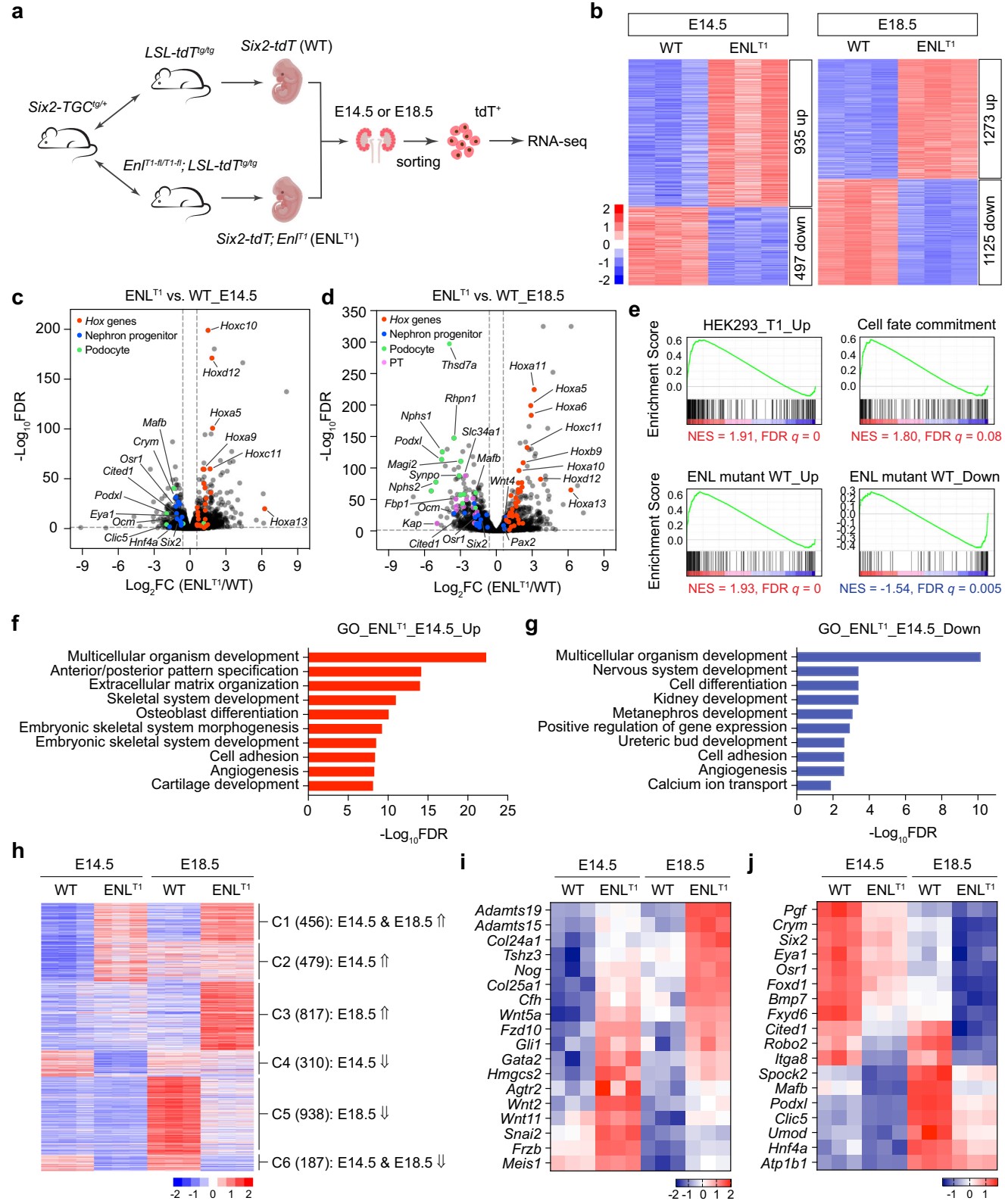

those identified in whole mutant kidneys (Fig. 3f, g, Supplementary Fig. 10c, d, and Supplementary Data 5).

Many of the DEGs are developmentally regulated; however, their expression changes in the ENL$^{WT}$ cells from E14.5 to E18.5 were not preserved in the ENL$^{T1}$ mutant cells (Supplementary Fig. 10e). To determine the pattern of gene expression changes in sorted ENL$^{T1}$ mutant cells *vs.* ENL$^{WT}$ cells, we grouped DEGs identified at both stages

into six clusters: C1, up in both, 456 genes; C2, up in E14.5 only, 479 genes; C3, up in E18.5 only, 817 genes; C4, down in E14.5 only, 310 genes; C5, down in E18.5 only, 938 genes; and C6, down in both, 187 genes (Fig. 3h, Supplementary Fig. 10f, and Supplementary Data 6, 7).

DEGs in cluster C1 were upregulated during the early stages in kidney development (E14.5) in ENL$^{T1}$ mutants and maintained a relatively high expression level at E18.5 compared to ENL$^{WT}$ kidneys.

**Fig. 3 | Gene expression changes in sorted ENL$^{TI}$ mutant cells from E14.5 and E18.5 Six2-ENL$^{TI}$ kidneys. a** Schematic of the mouse mating strategy and experimental design. Created in BioRender. Wen, H. (2025) https://BioRender.com/l62o789. **b** Heatmap representation of differentially expressed genes (DEGs) in sorted tdTomato$^+$ (tdT$^+$) cells from WT and Six2-ENL$^{TI}$ embryonic kidneys at E14.5 and E18.5 ($n = 3$). **c, d** Volcano plots of all expressed genes in the sorted tdT$^+$ kidney cells from E14.5 (**c**) and E18.5 (**d**) kidneys. The $x$-axis is the log$_2$ fold change (log$_2$FC) of counts per million (CPM) values from ENL$^{TI}$ vs. WT. The y-axis is $-$log$_{10}$ transformed FDR (log$_{10}$FDR) values of each gene. *Hox* genes and key marker genes of nephron progenitor, podocyte, and proximal tubule are highlighted in red, blue, green, and pink, respectively. **e** GSEA plots with tdT$^+$ Six2-ENL$^{TI}$ *vs.* tdT$^+$ WT kidney cells at E14.5 as ranking list and the indicated curated gene lists or GO term as gene sets. Positive enrichment scores are highlighted in red, while negative enrichment scores are in blue. **f, g** Enriched GO terms of the up (**f**) and down (**g**) DEGs identified in tdT$^+$ Six2-ENL$^{TI}$ cells at E14.5. The top ten GO biological process terms with FDR <0.05 are shown. **h** Heatmap of 3187 DEGs in sorted Six2-ENL$^{TI}$ cells from E14.5 and E18.5 kidneys grouped into six clusters. C1, upregulated at both E14.5 and E18.5; C2, upregulated at E14.5 only; C3, upregulated at E18.5 only; C4, downregulated at E14.5 only; C5, downregulated at E18.5 only; and C6, downregulated at both E14.5 and E18.5. Numbers of DEGs in each cluster are shown in parenthesis. **i, j** Heatmaps of key nephrogenesis genes in clusters C1 (**i**) and C6 (**j**). Color keys represent Z-score log$_2$CPM in **b**, **h–j**.

All *Hox* genes detected in embryonic kidneys were upregulated in sorted ENL$^{TI}$ mutant cells, with the majority of them grouped into the C1 cluster (Supplementary Fig. 10g), suggesting that the aberrant activation of *Hox* genes by mutant ENL occurs early and persists throughout kidney development. It is conceivable that the extensive and prolonged activation of *Hox* genes in ENL$^T$ mutant cells may cause further transcriptional alterations leading to impaired nephrogenesis. Besides *Hox* genes, several Wnt signaling pathway genes involved in the regulation of kidney development, such as *Wnt2*, *Wnt5a*, *Wnt11*, *Fzd10*, *Frzb*, and *Gli1*, also belong to cluster C1 (Fig. 3i). Additionally, this cluster includes many genes encoding transcription factors implicated in the development of various renal structures, including *Tshz3*, *Gata2*, *Hmgcs2*, *Snai2*, and *Meis1* (Fig. 3i). Overall, genes in cluster C1 likely account for the most significantly "gained" function of ENL$^{TI}$ mutant, as the top GO terms of DEGs in cluster C1 are almost identical to the GO terms of all up genes in sorted ENL$^{TI}$ mutant cells (Fig. 3f, Supplementary Fig. 10c, and Supplementary Data 6, 7).

Genes in cluster C6, downregulated in ENL$^{TI}$ mutant cells at both E14.5 and E18.5, are mostly associated with cell differentiation and metanephros development (Supplementary Data 7). The "stemness" genes of nephron progenitors, such as *Six2*, *Eya1*, *Crym*, *Osr1*, and *Cited1*, exhibit high levels of expression at early developmental stages and undergo downregulation along nephron differentiation and kidney development[59]. In Six2-ENL$^{TI}$ mutant cells, these genes were already significantly suppressed at E14.5, with further downregulation observed at E18.5 (Fig. 3j and Supplementary Data 6, 7). In contrast, marker genes for mature nephron structures are mainly expressed at later stages of normal development. While the expression levels of early podocyte and proximal tubule marker genes, such as *Mafb*, *Podxl*, *Clic5*, and *Hnf4a*, were already lower in ENL$^{TI}$ mutant cells compared to ENL$^{WT}$ cells at E14.5 (Fig. 3c, j), the majority of marker genes for podocyte and proximal tubule that were downregulated in ENL$^{TI}$ mutant cells belong to cluster C5, downregulated only at E18.5 (Fig. 3c, d, Supplementary Fig. 10f, h, i, and Supplementary Data 6, 7). Overall, these gene expression changes largely explain the phenotypes observed in mutant kidneys, which exhibit a lack of developed glomeruli and nephron tubules.

Interestingly, KEGG pathway analysis revealed that the gene set involved in pathways in cancer was enriched in DEGs from the sorted cells at E14.5 (Supplementary Fig. 10j, k and Supplementary Data 8). In addition, many upregulated DEGs identified in ENL$^{TI}$ mutant cells at E14.5 were enriched in signaling pathways, including cAMP, MAPK, Rap1, and Wnt signaling pathways, while genes in the PI3K-AKT pathway were up or down-regulated (Supplementary Fig. 10j, k). Together, these results indicate that the dysregulated gene expression in ENL$^{TI}$ mutant kidneys not only causes developmental defects, but may also create a predisposition to tumorigenesis.

### Spatial transcriptomic analysis of Six2-ENL$^{TI}$ kidneys

To investigate gene expression changes caused by Six2-ENL$^{TI}$ mutant in specific cell types within the context of tissue architecture, we employed the Visium HD Spatial Gene Expression platform, which enables whole-transcriptome spatial analysis at single cell-scale resolution with continuous tissue coverage. Integrated analysis of four ENL$^{WT}$ and four Six2-ENL$^{TI}$ mutant E18.5 kidneys identified 38 transcriptionally distinct clusters (Supplementary Fig. 11a–h). Clusters were annotated to specific cell types/anatomic renal structures based on marker genes[56,57], and clusters with the same annotation were merged, resulting in 20 distinct cell types/renal structures (Supplementary Fig. 11i, j). These include NPs (cap mesenchyme, marked by *Cited1$^+$* and *Six2$^+$*), committing NPs (PTA and RV, *Wnt4$^+$*), segments of S-shaped body (SSB-distal, *Irx1$^+$*; SSB-medial, *Osr2$^+$*; and SSB-proximal, *Ndnf$^+$*), elongated nephron tubules (early PT, *Hdc$^+$*; PT, *Slc34a1$^+$*; connecting tubule segment, *S100g$^+$*; DT, *Atp1b1$^+$*; and LOH, *Slc12a1$^+$*), cells in glomerulus (podocyte, *Podxl$^+$*; pericyte/mesangial cell, *Gja5$^+$*), stroma (capsular stroma, *Ntn1$^+$*; cortical stroma, *Ptn$^+$*; and medullary stroma, *Acta2$^+$*), ureteric bud (*Ret$^+$*), and collecting duct (*Plet1$^+$*). Additionally, vascular endothelial (*Fabp4$^+$*), blood vessel (*Rgs5$^+$*) and tissue-resident immune cell (*C1qa$^+$*) populations were also identified (Supplementary Fig. 11k and Supplementary Data 9).

Annotated cell type bins aligned closely with the H&E tissue images, effectively highlighting nephrogenesis defects identified through immunostaining in Six2-ENL$^{TI}$ mutants (Fig. 4a and Supplementary Fig. 11l). Six2-ENL$^{TI}$ mutants showed a smaller population of NPs but increased proportion of committing NPs and SSB segments (Fig. 4a, b), indicating reduced NP self-renewal, premature NP commitment and blockage of differentiation in nephrogenesis.

The overall gene expression changes detected through spatial transcriptomic analysis were largely consistent with those observed in bulk RNA-seq, and were mapped to specific cell types and renal structures. For instance, upregulated DEGs in Six2-ENL$^{TI}$ mutant NPs, committing NPs and nephrogenic SSB segments were all enriched in developmental processes, including multicellular organism development and anterior/posterior pattern specification (Fig. 4c and Supplementary Data 10). Notably, many *Hox* genes were not only upregulated in NPs and committing NPs, but also maintained at high levels in SSB and mature nephron cell types, where their expression should be downregulated during normal development (Fig. 4d and Supplementary Fig. 12a–c). In addition, we also identified unique processes enriched in specific cell types, such as kidney development and Wnt signaling pathway in Six2-ENL$^{TI}$ mutant NPs (Fig. 4c and Supplementary Fig. 12d, e). Wnt signaling plays an important role in regulating the balance between progenitor self-renewal and nephron differentiation. For example, *Wnt4* is normally not expressed in the self-renewal NPs within cap mesenchyme; rather, it is one of the earliest genes induced during progenitor commitment and plays a crucial role in this process[6]. We found that in Six2-ENL$^{TI}$ kidneys, *Wnt4* was upregulated in both NPs and committing NPs (Fig. 4d–f), supporting our hypothesis that aberrant activation of *Wnt4* in Six2-ENL$^{TI}$ mutant promotes premature NP commitment. Consistently, *Lef1*, a Wnt target gene, was also upregulated in ENL-mutant NPs and committing NPs (Supplementary Fig. 12e). Immunostaining further confirmed elevated LEF1 protein levels in CM, PTA, and RV (Supplementary Fig. 12h).

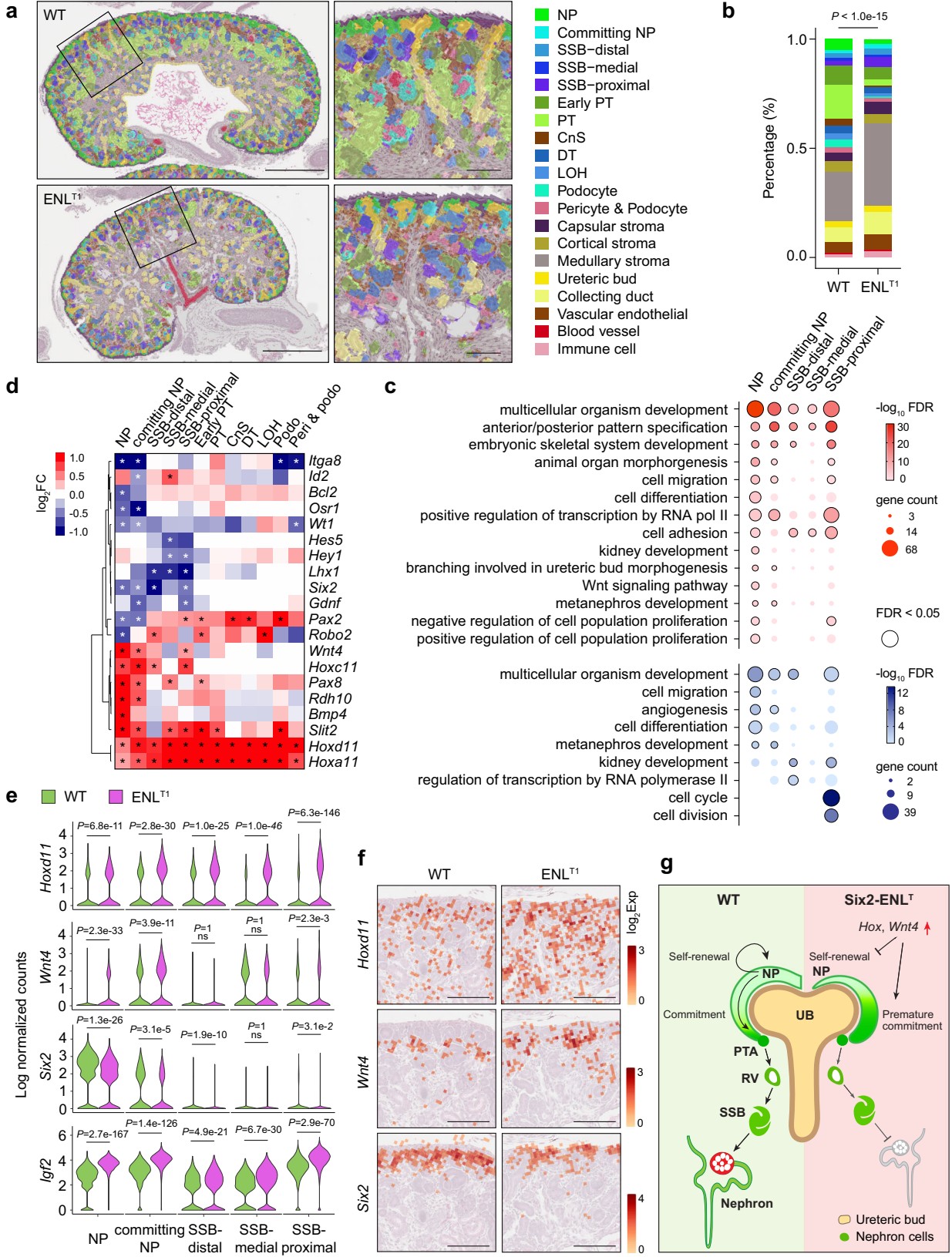

Additionally, other NP commitment markers, including *Pax8*, *Rdh10*, and *Clu*, were also upregulated in NPs and committing NPs in Six2-ENL^T1 mutant (Supplementary Fig. 12d, f, g). Furthermore, *Igf2*, an important fetal mitogen promoting cell proliferation and differentiation during kidney development, was upregulated in all nephron-related cell types (Fig. 4e).

Downregulated DEGs in Six2-ENL^T1 mutant NPs and committing NPs were enriched in developmental processes, such as multicellular organism development, angiogenesis, and metanephros development, while downregulated DEGs in distal and proximal SSB segments were enriched in kidney development (Fig. 4c, Supplementary Fig. 12d, and Supplementary Data 10). Many NP marker genes, including *Six2*, *Cited1*,

**Fig. 4 | Spatial transcriptomic analysis of gene expression changes in Six2-ENL[T1] mutants. a** Spatial distribution of annotated cell types and renal structures mapped to the H&E tissue section images. Areas outlined in the whole kidney images are enlarged in the zoom-in images on the right. NP nephron progenitor, SSB S-shaped body, PT proximal tubule, CnS connecting segment, DT distal tubule, and LOH loop of Henle. Images are representative of four pairs of kidney sections in the spatial transcriptomic analysis. Scale bars: 0.5 mm (whole kidney), and 0.1 mm (zoom-in). **b** Percentage of bins with annotated cell types as in (**a**) in WT and ENL[T1] tissue samples. Two-tailed Chi-square test. **c** Bubble plot showing enriched GO biological process terms for the upregulated (red) and downregulated (blue) DEGs in NP, committing NP, and SSB segments of Six2-ENL[T1] kidney sections. Color key represents $-\log_{10}$FDR values. Bubble size denotes gene counts. Black-circled bubbles indicate GO terms with FDR <0.05. **d** Heatmap of metanephros developmental genes across nephron cell types for DEGs identified in Six2-ENL[T1] NP, committing NP and SSB. The color key represents the $\log_2$FC of ENL[T1] vs. WT. DEGs with FDR <0.05 are marked with stars. **e** Violin plot showing expression levels (log normalized counts) of *Hoxd11*, *Wnt4*, *Six2*, and *Igf2* in NP, committing NP and SSB segments of WT and ENL[T1] samples. Wald test with adjusted *P* values shown; ns not significant. **f** Expression and spatial distribution of *Hoxd11*, *Wnt4*, and *Six2* in WT and ENL[T1] kidney sections. The color key represents $\log_2$ transformed UMI counts ($\log_2$Exp) in bins. *n* = 4 WT and 4 ENL[T1]. Scale bars, 0.1 mm. **g** Schematic model illustrating how ENL[T] mutations in the nephron lineage impair nephrogenesis.

---

*Osr1*, *Traf1*, *Meox2*, *Capn6*, *Spock2*, and *Itga8* were significantly downregulated in NPs and committing NPs, likely contributing to the reduced population of self-renewal NPs in Six2-ENL[T1] mutants (Fig. 4d–f, Supplementary Fig. 12d, f, g, and Supplementary Data 11). Notably, some PTA/RV marker genes, such as *Pax2*, *Tmem100*, *Snap91*, *Mycn*, and *Clmp*, were downregulated in mutant committing NPs (Fig. 4d and Supplementary Fig. 12d, f, g). Differentiating markers of nascent nephrons, such as *Lhx1*, *Bmp7*, *Aldh1a2*, *Emid1*, *Mafb*, *Mecom*, *Hes5*, and *Hey1*, were downregulated in mutant SSB segments (Supplementary Fig. 12d, f). Reduced protein levels of LHX1 were validated by immunostaining (Supplementary Fig. 12i). Meanwhile, some commitment and differentiating genes, including *Igfbp5*, *Sfrp2*, *Bmper*, and *Clu*, were elevated in mutant SSB segments and in more differentiated nephron cell types (Supplementary Fig. 12d, f, g). Collectively, these findings suggest that differentiation perturbations in Six2-ENL[T1] mutants occur at multiple stages of nephrogenesis.

Based on the observed phenotypes and gene expression changes, we propose a model for how ENL[T] mutants impact nephrogenesis in the nephrogenic lineage (Fig. 4g). ENL[T] mutants aberrantly activate *Hox* genes and NP commitment genes, such as *Wnt4* and *Pax8*, while downregulating NP "stemness" genes, such as *Cited1* and *Six2*, in the cap mesenchyme. This imbalance reduces NP self-renewal and promotes premature NP commitment. Additionally, some commitment genes (e.g., *Tmem100* and *Snap91*) are not fully activated in committing NPs, while others (e.g., *Sfrp2* and *Clu*) remain elevated in nascent nephrons, potentially disrupting the proper differentiation process. Downregulation of marker genes for different segments of nascent nephrons may further block or delay differentiation.

## ENL[T] mutants expressed in *Foxd1*+ stroma impair nephrogenesis

The nephron progenitors in cap mesenchyme are encapsulated by stromal fibroblasts, which also have essential functions during kidney morphogenesis. Stromal cells produce signals regulating nephron progenitor cell renewal, differentiation, ureter branching morphogenesis, and renal capsule formation[15,16]. Alterations, such as β-catenin activation, in the stroma can prevent nephron progenitor cell differentiation and result in histological and molecular features of human Wilms tumor, underscoring the critical role of stromal microenvironment in tumorigenesis[60]. FOXD1 is a well-known marker of stromal cells[12]. *Foxd1*+ stroma represents a multipotent self-renewing progenitor population that gives rise to stromal tissues within the interstitium, mesangium, and glomerular pericytes[61]. Therefore, to assess the impact of stromal ENL[T] mutations on nephrogenesis, we employed the *Foxd1*[GC] strain[62], which expresses an eGFP-Cre fusion protein from the *Foxd1* promoter/enhancer elements, enabling induced expression in the stromal compartment (Fig. 5a).

When breeding *Enl*[T-fl/+] mice with *Foxd1*[GC/+] mice, *Foxd1*[GC/+]; *Enl*[T1-fl/+] and *Foxd1*[GC/+]; *Enl*[T3-fl/+] pups with heterozygous expression of ENL[T] mutants were born at expected Mendelian ratio. However, these pups all died within hours after birth, with visibly smaller kidneys and urine-deficient bladders compared with their littermates (Supplementary Figs. 13a, b, 14a). Additionally, Foxd1-ENL[T] mutant kidneys exhibited reduced weight and size across developmental stages examined (E16.5, E18.5, and P0) compared to *Foxd1*[GC/+]; *Enl*[+/+] (ENL[WT]) counterparts (Fig. 5b and Supplementary Fig. 14b).

We performed H&E and immunofluorescence staining to evaluate developmental abnormalities in Foxd1-ENL[T] mutants. Structural changes of Foxd1-ENL[T] mutant kidneys were distinct from Six2-ENL[T] mutants (Fig. 5c–v and Supplementary Figs. 13c–o, 14c–v). TENASCIN staining revealed thickened capsular stroma in Foxd1-ENL[T] mutant across all stages (Fig. 5h, i, n and Supplementary Figs. 13i, j, 14h, i, n). This phenotype was further validated by staining of another stroma marker MEIS1/2 (Supplementary Fig. 13p). The mesenchymal domain capping ureteric bud also thickened and widened, as revealed by H&E, SIX2, and WT1/NCAM staining (Fig. 5c–g, o, p and Supplementary Figs. 13c–h, o, 14c–g, o, p). Ki-67 staining showed that Foxd1-ENL[T] mutants exhibited increased Ki-67+ cells in developing nephrons within the nephrogenic zone and in the capsular and interstitial stroma, suggesting that expression of ENL[T] mutants in the stromal domain induced hyperproliferation (Fig. 5w and Supplementary Figs. 13q, 14w).

Although the numbers of developing glomerulus were not significantly different between mutant and WT, the percentage of normal mature glomeruli was much lower in mutants (Fig. 5u, v and Supplementary Fig. 14u, v). Foxd1-ENL[T] mutants displayed a relatively normal WT1 distribution pattern in early nephron stages (Fig. 5f, g and Supplementary Figs. 13e–h, 14f, g), but their glomeruli exhibited various structural abnormalities, including cystic glomeruli, capillary tuft loss, reduced mesangial cells, dilated capillary loop, and capillary aneurysm, leading to dysfunctional glomeruli (Fig. 5c and Supplementary Figs. 13o, 14c). Cystic glomeruli were less prevalent in Foxd1-ENL[T] mutants compared to Six2-ENL[T] mutants. Further examination of PODXL and PDGFRB, markers for podocyte and mesangial cell, respectively, revealed a relatively normal distribution of podocytes in Foxd1-ENL[T] glomeruli (Fig. 5x and Supplementary Fig. 14x), differing from the nearly complete loss observed in Six2-ENL[T] mutants (Supplementary Fig. 8j, k). However, glomeruli in Foxd1-ENL[T] mutant had significantly reduced numbers of PDGFRB+ mesangial cells (Fig. 5x and Supplementary Fig. 14x), the central hub connecting and supporting glomerular structures to facilitate blood filtration and primary urine formation.

Nephron tubule development was also compromised in Foxd1-ENL[T] mutants, tubules were smaller with barely discernable lumens in Foxd1-ENL[T] mutants (Fig. 5c and Supplementary Figs. 13o, 14c). LTL and SLC12A3 staining revealed defects in tubular lumen development in Foxd1-ENL[T] kidneys, with both proximal and distal tubules reduced in number and exhibiting either no lumen space or only a very limited one (Fig. 5j–m, s, t and Supplementary Figs. 13k–n, 14j–m, s, t). Collectively, all these findings strongly suggest that the expression of ENL[T] mutants in the stromal compartment also has a deleterious impact on nephrogenesis.

## Gene expression changes in ENL[T1] mutant stroma lineage

To specifically investigate gene expression changes caused by ENL[T1] mutant expression in stroma lineage, we used the tdTomato (tdT) Cre

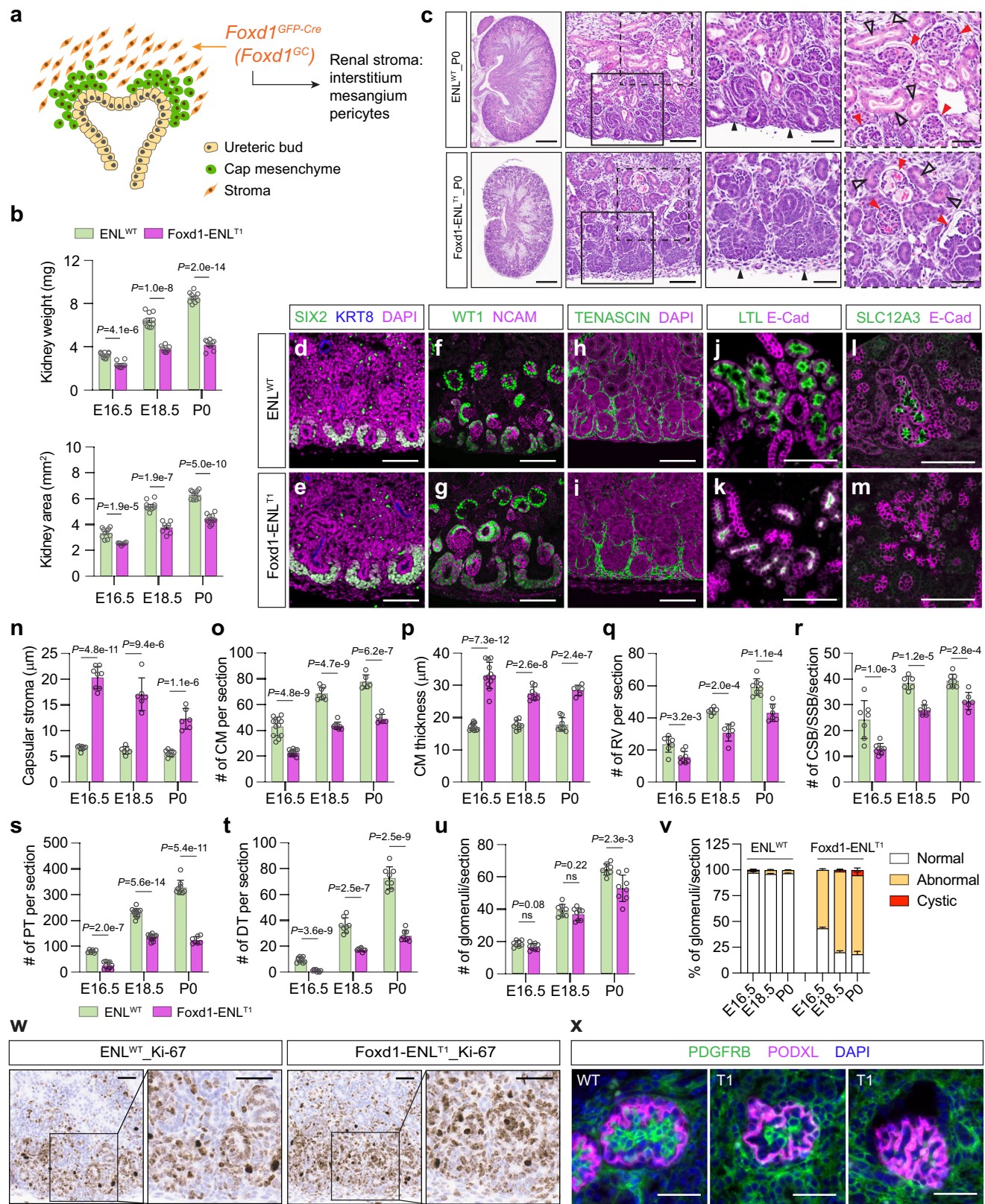

reporter to trace ENL[T1] mutant cells in embryonic kidneys. By crossing *Foxd1[GC/+]* mice with homozygous *LSL-tdT[tg]*; *Enl[T1-fl]* mice, we utilized FACS to isolate tdT[+] cells from E18.5 Foxd1-ENL[T1] kidneys and conducted RNA-seq analysis of the sorted cells (Fig. 6a). The tdT[+] cells isolated from *Foxd1[GC/+]; LSL-tdT[tg/+]* kidneys were used as ENL[WT] controls. Overall, the ratios of tdT[+] cells were comparable between ENL[WT] (29.23%) and ENL[T1] kidneys (28.09%) (Supplementary Fig. 15a),

indicating the total cell population of stroma lineage in ENL[T1] mutant kidneys was not significantly changed.

We identified 580 upregulated genes and 375 downregulated genes in tdT[+] cells from ENL[T1] mice relative to the ENL[WT] control mice (Fig. 6b, c and Supplementary Data 12). GSEA analysis unveiled that genes upregulated in HEK293 cells expressing human ENL[T1] and in *ENL*-mutant Wilms tumors were positively enriched in the sorted ENL[T1]

**Fig. 5 | Expression of ENL$^{TI}$ mutant in stromal compartment impairs nephrogenesis. a** Schematic of *Foxd1$^{GC}$* expression in nephrogenesis and cell types derived from *Foxd1$^+$* stromal progenitors. **b** Kidney weight and size. Sample sizes: E16.5, $n = 10$ ENL$^{WT}$ and 8 ENL$^{TI}$; E18.5, $n = 10$ ENL$^{WT}$ and 8 ENL$^{TI}$; and P0, $n = 10$ ENL$^{WT}$ and 10 ENL$^{TI}$. **c**, H&E staining of kidney sections from P0 ENL$^{WT}$ and Foxd1-ENL$^{TI}$ pups. Solid line boxes are magnified, focusing on the NGZ, and dash line boxes are magnified, focusing on glomeruli and PT. Red arrowheads indicate glomeruli; black open triangles indicate PT; and black triangles indicate capsular stroma. Scale bars: 0.5 mm (first column), 0.1 mm (second column), and 50 μm (last two zoom-in columns). **d–m** Immunofluorescence staining of SIX2 and KRT8 (**d, e**), WT1 and NCAM (**f, g**), TENASCIN (**h, i**), LTL and E-Cad (**j, k**), and SLC12A3 and E-Cad (**l, m**) in E18.5 kidney sections. Scale bars, 100 μm. **n–u** Quantification of nephron structures per section. Capsular stroma thickness (**n**) ($n = 8, 6, 8$ ENL$^{WT}$ and 8, 6, 6 ENL$^{TI}$); CM number (**o**) ($n = 12, 8, 6$ ENL$^{WT}$ and 12, 8, 6 ENL$^{TI}$); CM thickness (**p**) ($n = 12, 8, 8$ ENL$^{WT}$ and 12, 8, 6 ENL$^{TI}$); numbers of RV (**q**) and CSB-SSB (**r**) ($n = 7, 6, 8$ ENL$^{WT}$ and 8, 6, 6 ENL$^{TI}$), PT (**s**) ($n = 6, 12, 12$ ENL$^{WT}$ and 10, 12, 7 ENL$^{TI}$), DT (**t**) ($n = 10, 8, 8$ ENL$^{WT}$ and 8, 8, 8 ENL$^{TI}$), and glomeruli (**u**) ($n = 8$ per sample group per time point). **v** Percentage of glomeruli with normal, abnormal, and cystic morphologies in the same set of ENL$^{WT}$ and Foxd1-ENL$^{TI}$ kidneys as in (**u**). Data represent mean ± s.d.; two-tailed unpaired *t*-test; ns not significant. **w** Immunohistochemistry staining of Ki-67 in E18.5 ENL$^{WT}$ and Foxd1-ENL$^{TI}$ kidney sections. Scale bars, 50 μm. **x** Immunofluorescence staining of PDGFRB and PODXL in glomeruli of P0 kidneys. Scale bars, 40 μm. For (**w, x**), similar results were obtained in three independent experiments. Source data are provided as a Source Data file.

mutant FOXD1$^+$ progenitor-derived cells (Fig. 6d). GO analysis revealed that upregulated DEGs were enriched in development and cell adhesion-associated processes (Fig. 6e and Supplementary Data 13). Similar to Six2-ENL$^{TI}$ mutant cells, activation of *Hox* genes is a prominent feature in Foxd1-ENL$^{TI}$ mutant cells (Fig. 6c and Supplementary Fig. 15b). Additionally, several genes important for progenitor cell commitment and differentiation, including *Wnt4, Tmem100, Bmper, Frzb, Osr2,* and *Sfrp2*, were upregulated in the sorted mutant stromal cells (Supplementary Fig. 15c). Among the downregulated DEGs, kidney development was the most significantly enriched GO term (Fig. 6f, Supplementary Fig. 15d, and Supplementary Data 13). While a few stroma marker genes, such as *Vcam1, Nts, Meis2,* and *Fn1*, were upregulated, more stromal genes were downregulated (Fig. 6g). Some downregulated genes, such as *Pbx1, Tcf21, Pdgfrb, Ecm1,* and *Agtr2*, are known to be associated with aberrant nephrogenesis and ureteric branching when perturbed[11,63–68]. These findings suggest that ENL$^T$ mutants in the stromal lineage disrupt gene expression essential for proper nephrogenesis and kidney development.

Comparison of gene expression profiles of sorted ENL-mutant cells from E18.5 Foxd1-ENL$^{TI}$ and Six2-ENL$^{TI}$ kidneys revealed 185 genes upregulated and 104 downregulated in both lineages (Fig. 6h and Supplementary Data 14). These shared DEGs include *Hox* genes and transcription regulators involved in pattern specification, cell differentiation, and kidney development, suggesting a common core set of target genes impacted by ENL mutations in both lineages (Supplementary Fig. 15e).

### Spatial transcriptomic analysis of Foxd1-ENL$^{TI}$ kidneys

To understand how ENL$^{TI}$ mutation in the stromal compartment impedes nephrogenesis, we performed spatial transcriptomic analysis using the Visium HD platform. Integrated analysis of four E18.5 kidneys identified 32 transcriptionally distinct clusters that were merged into 19 annotated cell types or anatomic kidney structures (Fig. 7a and Supplementary Fig. 16a–l). Consistent with histology and immunostaining results, we observed an expansion of NP and committing NP populations, along with a moderately expanded capsular stroma (Fig. 7a, b and Supplementary Fig. 16m). Spatial transcriptome profiling revealed that ENL$^{TI}$ expression in stroma lineage primarily affected local gene expression, with a higher number of DEGs in the three stroma compartments (capsular, cortical, and medullary stroma) than other renal cell types (Supplementary Data 15). GO analysis showed that upregulated DEGs in the stroma were enriched in angiogenesis, cell adhesion, development, cell migration, and gene transcription (Fig. 7c, Supplementary Fig. 17a–d, and Supplementary Data 16), many of which were also observed in bulk RNA-seq of sorted Foxd1-ENL$^{TI}$ mutant cells. Notably, upregulation of *Hox* genes was confined to stromal cells expressing mutant ENL (Fig. 7d). While no GO terms were significantly enriched in down DEGs in stroma, we did observe downregulation of key regulators involved in kidney development and cell differentiation in the stromal lineage (Supplementary Fig. 17e, f).

One interesting observation in the Foxd1-ENL$^T$ kidneys was the expansion of NP and committing NP populations. As mutant ENL in Foxd1-ENL$^{TI}$ kidneys primarily affects gene expression in the stroma lineage, it is conceivable that nephron progenitors receive signals from ENL$^T$ mutant stromal cells through stroma-nephron interactions, promoting cap mesenchyme proliferation while hindering proper nephron differentiation. To explore the potential paracrine ligand-receptor (L-R) interactions that contribute to nephrogenesis defects in Foxd1-ENL$^{TI}$ kidneys, we performed cell-cell communication analysis using LIANA+[69,70]. Among the significant and specific co-expressed L-R pairs identified, we focused on L-R interactions involving ligands differentially expressed between ENL$^{WT}$ and Foxd1-ENL$^{TI}$ mutant kidneys in the nephrogenic zone (NGZ) stroma, including the capsular and cortical stroma. Several ligand-encoding genes, including *Fn1, Thbs1, Col4a2, Igf2,* and *Igfbp4*, were upregulated, while others, including *Gpc3* and *Ntn1*, were downregulated in the stroma of Foxd1-ENL$^{TI}$ mutants (Supplementary Fig. 18a, b). Importantly, the expression of these ligands and their engaged receptors (e.g., *Fn1-Itga8, Igfbp4-Lrf6, Gpc3-Cd81,* and *Ntn1-Unc5c*) localized in adjacent domains within the stroma and NPs, respectively (Fig. 7e and Supplementary Fig. 18c–e). Increased protein levels of fibronectin (FN) and its localization adjacent to ITGA8 in cap mesenchyme were also confirmed by immunostaining (Fig. 7f). Consistent with the up- or downregulation of these ligands in stroma, L-R interactions were correspondingly increased or decreased in ENL$^{TI}$ mutants (Fig. 7g and Supplementary Fig. 18f–k), potentially affecting both stroma-NP and stroma-UB interactions. Together, these findings suggest that likely altered paracrine L-R interactions between stroma, nephron, and ureteric epithelium collectively contribute to impaired nephrogenesis in Foxd1-ENL$^{TI}$ kidneys (Fig. 7h).

## Discussion

Throughout development, epigenetic mechanisms play crucial roles in the precise regulation of gene expression to establish proper cell fate commitment and direct terminal differentiation in tissue specification. Mutations of epigenetic regulators or dysregulation of epigenetic landscapes are common causes of developmental disorders. Disruptions to the finely tuned gene expression network in development can impair cell fate determination and lead to cancers. One such example is Wilms tumor, the most common type of kidney cancer in children. Wilms tumor is a prototypical embryonic malignancy arising from the failure of embryonic nephrogenic cells to undergo terminal differentiation[7]. While the exact cause of Wilms tumor is not fully understood, recent genomic analyses have uncovered a series of mutations in genes encoding epigenetic regulators, underscoring the critical role of proper epigenetic and transcriptional regulation in kidney development[19,20]. Among these mutations, hotspot mutations in the ENL YEATS domain are the most frequent, accounting for 4–9% of Wilms tumor cases. However, despite this prevalence, animal models for these hotspot ENL mutations had not been established at the inception of this project, hindering our understanding of these

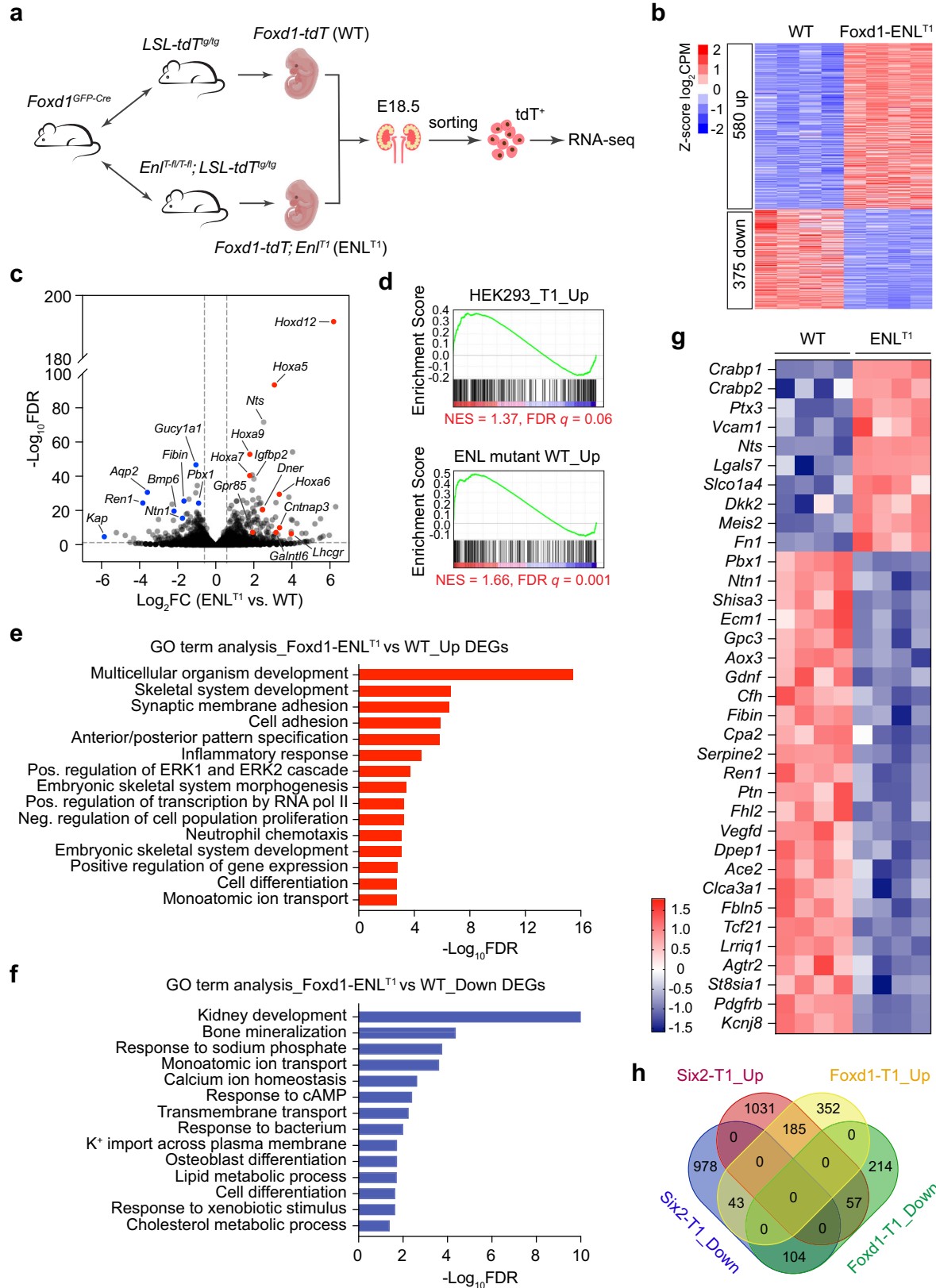

cancer-associated mutations in normal kidney development and disease pathogenesis.

In this study, we have generated Cre/loxP-based cKI mouse models mimicking two prototype ENL YEATS domain mutations found in Wilms tumor: ENL[T1], representing the most prevalent insertion mutation, and ENL[T3], representing a deletion mutation. By utilizing three Cre lines, including the ubiquitous *CMV-Cre[tg]* and tissue-specific *Six2-TGC[tg]* and *Foxd1[GC]*, we have demonstrated that either whole-body or kidney-specific expression of ENL tumor mutations impairs kidney development, resulting in smaller kidneys with severe structural defects and neonatal lethality. Similar developmental defects have been observed in a recent study using a *Wt1[GFP-Cre]* driven mouse model,

**Fig. 6 | Gene expression changes in sorted ENL$^{TI}$ mutant stromal cells from E18.5 Foxd1-ENL$^{TI}$ kidneys. a** Schematic of the mouse mating strategy and experimental design. Created in BioRender. Wen, H. (2025) https://BioRender.com/l62o789. **b** Heatmap representation of differentially expressed genes (DEGs) in sorted tdTomato$^+$ (tdT$^+$) cells from WT and Foxd1-ENL$^{TI}$ embryonic kidneys at E18.5 ($n = 4$). **c** Volcano plots of all expressed genes in the sorted tdT$^+$ stromal cells from E18.5 kidneys. The $x$-axis is the log$_2$FC of CPM values from ENL$^{TI}$ vs. WT. The y-axis is $-$log$_{10}$ transformed FDR values of each gene. Representative *Hox* genes and genes

upregulated in human Wilms tumors with ENL$^T$ mutations are highlighted in red, and downregulated stromal genes are highlighted in blue. **d** GSEA plots with tdT$^+$ Foxd1-ENL$^{TI}$ vs. tdT$^+$ WT kidney stroma cells as ranking list and the indicated curated gene lists as gene sets. **e, f** Enriched GO terms (FDR <0.05) of upregulated (**e**) and downregulated (**f**) DEGs in tdT$^+$ Foxd1-ENL$^{TI}$ cells. **g** Heatmap of stroma marker genes differentially expressed in sorted tdT$^+$ WT and Foxd1-ENL$^{TI}$ cells. The color key represents Z-score log$_2$CPM. **h** Venn diagram of overlapping DEGs in Six2-ENL$^{TI}$ nephron cells and Foxd1-ENL$^{TI}$ stroma cells from E18.5 kidneys.

in which ENL$^{TI}$ mutant is expressed in both nephron and stromal lineages[71]. Our study demonstrates that expression of ENL$^{TI}$ mutant in either the nephrogenic or stromal lineage is sufficient to impede kidney development. Moreover, within the same cell lineage, similar phenotypes have been observed in mice expressing ENL$^{TI}$ and ENL$^{T3}$, suggesting that insertion and deletion mutations in the ENL YEATS domain function through a similar mechanism.

Histologically, the phenotypes observed in Six2-ENL$^T$ mutants are more severe compared to Foxd1-ENL$^T$ mutants, indicating impairment caused by mutating ENL in nephrogenic cells is dominant. For instance, the Six2-ENL$^T$ kidneys are absent of normal mature glomeruli, substituted with cystic glomeruli, whereas the Foxd1-ENL$^T$ mutant kidneys have a normal number of developing glomeruli, the majority of which show abnormal glomerular structures with only a few cystic glomeruli observed. In addition, cap mesenchyme in the Six2-ENL$^T$ mutants appears thinner than the wildtype counterpart, whereas in the Foxd1-ENL$^T$ mutant kidneys, both cap mesenchyme and capsular stroma are expanded. These observations collectively suggest that expression of ENL YEATS domain tumor mutations in nephrogenic and stromal lineages impedes nephrogenesis through different pathways.

ENL functions as a transcriptional coactivator by recruiting the SEC and DOT1L complex regulating transcription elongation[40,43]. The developmental defects in ENL$^T$ mutant kidneys are likely attributed to alterations in gene expression, particularly activation of the *Hox* gene clusters. Posterior *Hox* genes, such as *Hox9* to *Hox11*, play pivotal roles in kidney development regulation. Disruption of *Hox9*, *Hox10*, and *Hox11* functions leads to cellular-level lineage infidelity in the kidney[42]. *Hox10* is involved in the formation of the stromal compartment[14]. Homozygous mutation of *Hoxa11* and *Hoxd11* results in dramatically hypoplastic kidneys with defects in ureteric bud branching morphogenesis, while mutation of all three *Hox11* paralogs completely blocks the initial stage of kidney formation[41]. In Six2-ENL$^T$ and Foxd1-ENL$^T$ mutant kidneys, the majority of *Hox* genes are upregulated, and the aberrant activation can be seen as early as E14.5 in Six2-ENL$^T$ cells. Through ChIP-seq analysis in cultured cells, we and others have previously demonstrated that *HOX* genes are direct targets of the wildtype and mutant ENL[43,72]. ENL$^T$ mutants activate *HOX* gene expression through aberrant accumulation on HOX gene promoters[43,72]. Notably, dysregulation of *HOX* expression is a key gene signature in Wilms tumors with ENL mutations[20]. Furthermore, upregulated DEGs identified in ENL-mutant Wilms tumors are overall positively enriched in the Six2-ENL$^T$ and Foxd1-ENL$^T$ mutant kidneys, highlighting the clinical relevance of our mouse models.

The impaired nephrogenesis in the Six2-ENL$^T$ mutant kidneys is likely attributable, at least in part, to the early activation of Wnt signaling and reduced expression of nephron progenitor "stemness" genes, such as *Six2*. SIX2 is a well-known key transcription factor involved in nephrogenesis, regulating the self-renewal and maintenance of the nephron progenitor population during kidney development. Targeted deletion of *Six2* in nephron progenitors derepresses the differentiation program, triggering premature nephrogenesis and resulting in nephron progenitor depletion and reduced nephron formation[49]. SIX2 controls nephron progenitor differentiation and nephron formation in part by modulating the expression of *Wnt4*. Wnt signaling is the driving force for nephron progenitor cell

differentiation[6]. *Six2*$^+$ nephron progenitors are "primed" for differentiation in response to inductive Wnt signaling. Induction of *Wnt4* in nascent nephrons, such as renal vesicles, requires low SIX2 and high β-catenin levels[6,73]. In the Six2-ENL$^T$ mutant kidneys, upregulation of *Wnt4* coincides with reduced levels of SIX2 in cap mesenchyme, resulting in premature commitment of nephron progenitors. Consistent with our findings, *Wnt4* is also upregulated in NPs of Wt1-ENL$^{TI}$ mutant kidneys[71].

Additionally, the downregulation of several other nephron progenitor "stemness" genes, including *Cited1*, *Crym*, *Eya1*, and *Osr1*, along with the upregulation of certain nascent nephron marker genes, such as *Bmper*, *Tmem100*, *Pax8*, *Sfrp2*, and *Hoxa11*[56], further indicates the premature commitment of nephron progenitors in the Six2-ENL$^T$ mutant kidneys. Despite the widespread gene expression changes, the lack of an antibody recognizing mouse ENL at endogenous levels makes it challenging to distinguish direct from secondary effects of ENL mutations in mouse models. Our previous studies in human cells have identified *HOX* genes as direct targets of ENL mutants[43]. Additionally, HOX11 paralogs, especially HOXD11, have been reported to bind a distal enhancer of the *Wnt4* gene to regulate its expression in mice[74]. Future studies are needed to elucidate the hierarchy of gene expression changes driven by ENL mutations.

To date, the specific cells of origin for Wilms tumor are not definitively established. Previous studies have shown that overexpression of Lin28 in mice significantly expands nephron progenitors by blocking their final wave of differentiation, leading to a pathology highly reminiscent of Wilms tumor[75]. Tumor formation occurs only when Lin28 is aberrantly expressed in multiple derivatives of intermediate mesoderm, implicating a multipotential renal progenitor as the cell of origin[75]. Another study suggests that nephron progenitors, but not stromal progenitors, give rise to Wilms tumors in mouse models with β-Catenin activation or *Wt1* ablation and *Igf2* upregulation[76]. However, activating mutations of β-Catenin are present in both the stroma and blastema parts of Wilms tumor[60]. In line with this, activation of β-Catenin in the stromal lineage non-autonomously prevents the differentiation of nephron progenitors and results in histological and molecular features of human Wilms tumor, underscoring the important role of stromal microenvironment in tumorigenesis[60].

Wilms tumors harboring ENL mutations are usually triphasic, consisting of blastema, epithelial, and stromal components[19,20]. In the current study, expression of ENL$^T$ mutants in the *Foxd1*$^+$ stromal compartment also leads to impaired nephrogenesis and neonatal lethality, further demonstrating the importance of stroma-epithelium crosstalk during development[15,16]. The cap mesenchyme expansion observed in Foxd1-ENL$^{TI}$ kidneys resembles a phenotype seen in mice lacking stromal progenitors[15]. This expansion has been linked to the loss of the FAT4 cadherin in cortical stroma and disruptions in FAT4-mediated cell-cell interactions through YAP/TAZ or DCHS1 signaling[77–80]. However, we did not detect transcriptional change of *Fat4* in Foxd1-ENL$^{TI}$ mutant kidneys. Instead, we identified several ligand-encoding genes, including *Fn1, Thbs1, Col4a2, Gpc3*, and *Ntn1*, that were either up- or downregulated in the ENL-mutant stroma. Previous studies have shown that these ligands are important for kidney development, and their dysregulation leads to various developmental defects. For instance, deletion of *Fn1* in cultured

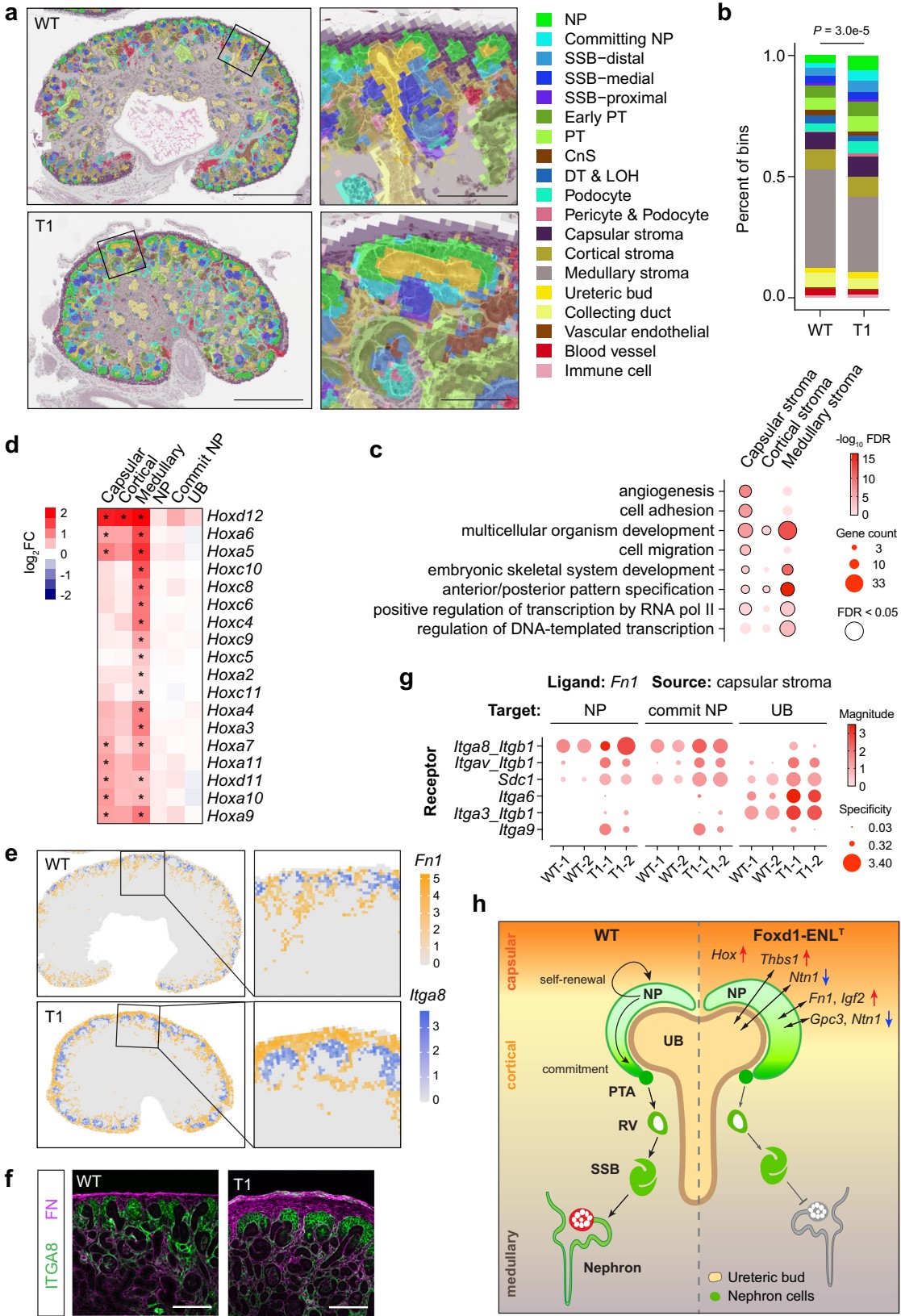

metanephric mouse kidneys reduces UB epithelial branching and kidney size[81]. Similarly, *Gpc3*-null mice display renal branching defects and kidney dysplasia[82,83], while conditional knockout of *Ntn1* in stroma results in hypoplastic kidneys with extended nephrogenesis[84,85]. Our study suggests that the expression of ENL^T mutants in the stroma leads to alterations in the interactions between these stromal ligands with

their receptors in both the NP and UB compartments. It would be interesting to investigate in the future how dysregulation of stroma-epithelium crosstalk through these ligand-receptor pairs contributes to developmental defects.

Overall, in this study, through integrated genetic mouse modeling, phenotype characterization, and transcriptomic analyses, we have

**Fig. 7 | Spatial transcriptomic analysis of gene expression changes in Foxd1-ENL[T1] mutants. a** Spatial distribution of annotated cell types and renal structures mapped to the H&E tissue section images. Areas outlined in the whole kidney images are enlarged in the zoom-in images on the right. NP nephron progenitor, SSB S-shaped body, PT proximal tubule, CnS connecting segment, DT distal tubule, and LOH loop of Henle. Images are representative of two pairs of kidney sections in the spatial transcriptomic analysis. Scale bars: 0.5 mm (whole kidney), and 0.1 mm (zoom-in). **b** Percentage of bins with annotated cell types as in (a) in WT and ENL[T1] tissue samples. Two-tailed Chi-square test. **c** Bubble plot showing enriched GO biological process terms for upregulated DEGs in the capsular, cortical, and medullary stroma of Foxd1-ENL[T1] kidney sections. Color key represents −log[10]FDR values. Bubble size denotes gene counts. Black-circled bubbles indicate GO terms with FDR <0.05. **d** Heatmap representation of *Hox* gene

expression changes across stroma, NP, committing NP, and UB clusters in Foxd1-ENL[T1] samples. The color key represents the log[2]FC of ENL[T1] vs. WT. DEGs with FDR <0.05 are marked with stars. **e** Spatial expression and distribution of *Fn1* in nephrogenic zone stroma and *Itga8* in NP/committing NP clusters in WT and ENL[T1] tissue samples. The color key represents log normalized counts. **f** Immunofluorescence staining of FN and ITGA8 in E18.5 WT and Foxd1-ENL[T1] kidney sections. Results are representative of three independent experiments. Scale bar, 100 μm. **g** Bubble plot depicting the strength and specificity of paracrine ligand-receptor interactions between capsular stroma *Fn1* and its engaged receptors in NP, committing NP, and UB in WT and ENL[T1] samples. The color key represents log magnitude *P* values. Bubble size denotes log specificity *P* values. Robust rank aggregation was used for *P* value calculation. **h** Schematic model illustrating how ENL[T] mutations in the stromal lineage impair nephrogenesis.

delineated impaired nephrogenesis with distinct phenotypes upon expression of ENL YEATS mutants in nephrogenic and stromal lineages, and uncovered associated transcriptional changes underlying the deficits in kidney development. Our study not only sheds light on how ENL YEATS domain mutations impede nephrogenesis through different pathways in nephrogenic and stromal lineages, but also paves the ways for further investigation into their roles in tumorigenesis. However, there are some limitations of our study. Although the ENL[T] mutant kidneys share certain gene expression features with human Wilms tumors with *ENL* mutations, so far tumor formation has not been observed in our mouse models. The lack of tumorigenesis in our current mouse models may be attributed to the severe defects in kidney development caused by ENL[T] mutations across lineages. Mutant pups succumbed soon after birth due to nonfunctional kidneys, precluding the time window necessary for tumor formation, which typically spans weeks to months. In the future, the controlled induction of ENL YEATS domain mutations in subpopulations within one or more lineages during embryonic development may offer a promising avenue for studying their role in tumorigenesis. Additionally, it would be valuable to explore the combination of ENL mutations with other concurrent but rare alterations found in human patients[19,20], such as *Ctnnb1* mutations and biallelic *Igf2* expression due to loss of imprinting, using the ENL[T] mouse models.

## Methods

This research complies with all relevant ethical regulations of the Van Andel Institute that approved the study protocol.

### Animal experiments

All animal experiments were approved by and performed in accordance with the guidelines of the Institutional Animal Care and Use Committee (IACUC) at the Van Andel Institute. Mice were housed in a specific pathogen-free facility with controlled temperature (-72 °F) and humidity (30–70%). They were maintained under a 12-h light/dark cycle, with lights on at 7:00 AM and off at 7:00 PM. *CMV-Cre* stain (*B6.C-Tg(CMV-cre)1Cgn/J*, JAX stock # 006054)[44], *Six2-TGC[tg]* BAC transgenic strain (*Tg(Six2-EGFP/cre)1Amc/J*, JAX stock # 009606)[4], *Foxd1[GC]* strain (*B6;129S4-Foxd1[tm1(GFP/cre)Amc]/J*, JAX stock # 012463)[62] and the Rosa26 tdTomato reporter *LSL-tdT[tg]* strain Ai14D (B6.Cg-*Gt(ROSA)26Sor[tm14(CAG-tdTomato)Hze]/J*, JAX Stock # 007914)[58] were purchased from the Jackson Laboratory. *Foxd1[GC]* strain contains the *Foxd1[GFPCre]* knock-in allele, which abolishes *Foxd1* gene function. *Six2-TGC[tg]* BAC transgenic strain has recently been reported to exhibit aberrant *Six3* expression in nephron progenitor cells driven by a distal enhancer of *Six2*[48]. *Six2-TGC[tg]* and *Foxd1[GC]* alleles were backcrossed to C57BL/6J mice for more than eight generations to get them close to congenic to C57BL/6J.

The conditional knock-in mouse models for *Mllt1[T1-fl]* (*ENL[T1-fl]*) and *Mllt1[T3-fl]* (*ENL[T3-fl]*) were generated in C57BL/6J background at the Jackson Laboratory using CRISPR/Cas9-mediated editing and homologous recombination. Targeting strategy was shown in Supplementary Fig. 1a. Four gRNAs used to induce DNA double strand breaks are

provided in Supplementary Data 17. The conditional loxP cassette included the wildtype *Enl* exon 4, coding sequence of *Enl* exons 5–12, a BGH polyadenylation signal followed by a mutant *Enl* exon 4 encoding a mutant ENL (ENL[T1] or ENL[T3]) between the 5′ and 3′ loxP sites. The conditional mutant exon 4 has either a 9-nt insertion (AACCACCTG) encoding a duplication of p.117_118NHL (p.117_118insNHL, ENL[T1]) or a 6-nt deletion plus 1-nt mutation (from AACCCTCCT to AAA) to replace p.111_113NPP with K (p.111_113NPP > K, ENL[T3]). This loxP cassette was flanked by a 2.8 kb left homologous arm (LHA) and a 3.0 kb right homologous arm (RHA) of the mouse *Enl* gene in the donor construct. Short- and long-range genomic PCRs, and DNA Sanger-sequencing were performed to confirm accurate knock-ins. Primers used for genotyping are provided in Supplementary Data 17. Germline transmission was confirmed in founder lines and backcrossed to C57BL/6 J mice for more than eight generations before being used in experiments. Both male and female mice were included in this study.

### Fibroblast cell culture and virus transduction

0.5 cm long mouse tail was taken from mice younger than 1 month of age and washed in PBS (Corning, #21040CM) containing 500 U/mL penicillin/ streptomycin (Corning, #30-009-CI) three times. About 1 mL of 1 mg/mL collagenase D (Roche, #11088866001) in complete DMEM medium (Corning, #10-017-CV, supplemented with 10% fetal bovine serum, 2 mM L-glutamine and 100 U/mL penicillin/streptomycin) was used to digest minced mouse tail in $CO_2$ incubator at 37 °C overnight. Tail fibroblast cells were filtered with a 40-μm cell strainer and centrifuged at 500×*g* for 5 min to collect cells. The cell pellet was resuspended and cultured in a DMEM medium. Lentivirus expressing CMV-Cre was packaged in human embryonic kidney 293T (HEK293T) cells, using X-treme GENE HP DNA transfection reagent (Roche, #06366546001) in accordance with the manufacturer's instructions. The virus was collected 48 h after transfection and applied to mouse tail fibroblast cells for transduction. Transduced mouse fibroblasts were selected with 2 μg/mL puromycin (Gibco, #A1113803) and collected for RNA extraction. HEK293T cells (ATCC CRL-3216) were maintained in DMEM medium and tested mycoplasma negative.

### Kidney sample preparation

Timed mating was set up to collect kidneys at E14.5, E16.5, E18.5, and P0. *CMV-Cre*, *Six2-TGC[tg]*, or *Foxd1[GC]* Cre strains were crossed with *Enl[T-fl]* mice. Kidneys or intact urinary system were collected for phenotypic analysis. Collected kidneys were weighted and imaged under Zeiss Discovery V12 microscope. Size of kidney was measured with Fiji (ImageJ2) software in longitudinal section which gives the largest area.

### H&E and immunostaining of kidneys

Collected mouse kidneys were fixed with 10% neutral buffered formalin (VWR, #1600128) for 24 h. All kidneys were processed, embedded, sectioned, and stained at the Pathology and

Biorepository Core (PBC) at Van Andel Institute. Standard Hematoxylin and Eosin (H&E) staining, immunochemistry (IHC) and immunofluorescence staining were also performed at the PBC core. For IHC staining, kidney sections were incubated with primary antibodies: anti-WT1 (rabbit, Abcam #ab89901, 1:300 dilution), anti-E-Cad (rabbit, Cell signaling #3195S, 1:200 dilution) or anti-Ki-67 (rabbit, Abcam #ab16667, 1:100 dilution) before developing with DAB chromogen. Sections with IHC staining was incubated with hematoxylin for 5 min at last. All sections were scanned by Leica Aperio AT2 scanner at 20x. For immunofluorescence staining, kidney sections were incubated with primary antibodies: anti-Six2 (rabbit, MyBioSource #MBS7604120, 1:200 dilution), anti-WT1 (rabbit, Abcam #ab89901, 1:300 dilution), Fluorescein-LTL (Lotus Tetragonolobus Lectin #F1321, 1:200 dilution), anti-KRT8 (rat, DSHB #TROMA-I-S, 1:50 dilution), anti-E-Cad (mouse, BD Biosciences #610181, 1:200 dilution), anti-PODXL (mouse, R&D systems, #MAB1556, 1:200 dilution), anti-PDGFR (rabbit, Abcam #ab32570, 1:100 dilution dilution), anti-TENASCIN (rabbit, Sigma #AB19011, 1:100 dilution), anti-SLC12A3 (rabbit, Abcam #ab95302, 1:100 dilution), anti-NCAM (mouse, Sigma #C9672, 1:50 dilution), anti-LEF1 (rabbit, Cell signaling #2630S, 1:250 dilution), anti-LHX1 (rabbit, Abcam #ab229474, 1:100 dilution), anti-MEIS1/2/3 (mouse, Active Motif #39096, 1:100 dilution), anti-FN (rabbit, Proteintech #15613-1-AP, 1:200 dilution), anti-ITGA8 (goat, R&D #AF4076-SP, 1:100 dilution) and detected by the secondary antibodies: Alexa Flour 488 goat anti-rabbit (Invitrogen, #A11034), Alexa Flour 488 goat anti-mouse (Invitrogen, #A11001), Alexa Flour 488 donkey anti-goat (Invitrogen, #A11055), Alexa Flour 546 goat anti-rabbit (Invitrogen, #A11035), Alexa Flour 546 goat anti-mouse (Invitrogen, #A11030), Alexa Flour 546 goat anti-rat (Invitrogen, #A11081). Sections were stained with DAPI (Sigma, #D9542) before mounting with Vectashield Mounting Medium (Vector labs, #H-1000). Sections were scanned by Zeiss Axioscan 7 scanner or imaged by Nikon A1 plus-RSi laser scanning confocal microscope. Images were processed and analyzed using Fiji (ImageJ2) and QuPath. Antibodies used in this study are provided in Supplementary Data 18.

## Nephron structure quantification

Immunofluorescence-stained images of longitudinal kidney sections with maximal surface area were used for quantification. Kidney sections with Six2 staining were used for quantification of cap mesenchyme (CM). Kidney sections with NCAM and WT1 staining were used for the quantification of RV, CSB/SSB, and glomerulus. The proximal tubule and distal tubule were quantified by LTL/E-Cad and SLC12A3/E-Cad staining, respectively. Kidney sections with TENASCIN staining were used for the quantification of capsular stroma. Number of cap mesenchyme (CM), CSB and SSB, proximal tubule, distal tubule, and glomerulus was quantified per section. The thickness of CM, nephrogenic zone, and capsular stroma was measured at five different locations within sections. At least five individual kidneys were quantified for Cre[+] control and ENL[T] at each time point.

## RNA extraction and semiquantitative RT-PCR analyses

Total RNA was isolated using the RNeasy Plus Mini kit (Qiagen, #74134) or RNeasy Plus Micro kit (Qiagen, #74034) and reverse transcribed with the iScript cDNA Synthesis Kits (Bio-Rad, #1708840) in accordance with the manufacturer's instructions. Expression of *Cre*, wild-type *Enl* and *Enl[T]* mutant was detected by RT-PCR followed by DNA agarose electrophoresis. *Gapdh* was used as an internal control. The expression level of wildtype and ENL[T] mutant was compared by semiquantitative RT-PCR using wildtype and mutant-specific primers, followed by SDS-PAGE electrophoresis. RT-PCR strategy was shown in Supplementary Fig. 2b. Primers used for RT-PCR are provided in Supplementary Data 17.

## Fluorescent-activated cell sorting of tdT[+] kidney cells

Timed mating was set up by crossing homozygous *LSL-tdT[tg/tg]* or *Enl[T1-fl/T1-fl]*; *LSL-tdT[tg/tg]* mice with *Six2-TGC[tg/+]* or *Foxd1[GC/+]* mice to collect embryonic kidneys at E14.5 and E18.5. The kidney was dissociated as previously described[4]. E14.5 kidneys were minced and treated with 200 μL 0.25% trypsin (Gibco, #25200-056) at 37 °C for 3 to 5 min. About 600 μL complete DMEM medium was added to stop the reaction. E18.5 kidneys were minced and incubated with 1 mL digestion solution containing 2 mg/mL collagenase II at 37 °C for 1 h. Cells were filtered with a 40-μm cell strainer and centrifuged at 500×g for 5 min to collect cell pellets. The cell pellet was resuspended in sorting buffer (HBSS with 25 mM HEPES, 2 mM EDTA, and 2% FBS) for sorting on BD FACSymphony S6. DAPI (4′,6-diamidino-2-phenylindole) was added before sorting. The gating strategy for cell sorting is included in the Source Data file.

## Bulk RNA-seq and analysis

RNA was isolated as described above, and then PolyA plus RNA-seq sample libraries were prepared using the KAPA Stranded mRNA-Seq Kit v7.21 (Roche, #KK8421) and the SMART-seq v4 Ultra Low Input RNA kit (Takara Bio USA, #634888) (used for sorted tdT[+] cells) following the standard protocol. RNA-seq samples were sequenced by Illumina NovaSeq 6000 for 2 × 50 bp (Six2-ENL[T]) and 2 × 75 bp (Foxd1-ENL[T]) paired-end reads. Raw reads in Fastq files were mapped to the mouse genome (mm10) by HISAT2 (v2.1.0)[86] with -k 1. Bam files were converted by Samtools (v1.10). Gene reads count tables were generated by HTSeq (v0.11.3)[87] with −stranded=no -a 0. CPM (counts per million) and fold change values were calculated edgeR (v3.16.5)[88] with trimmed mean of *M*-values (TMM) and exact test model. Differentially expressed genes (DEGs) between ENL[T] mutant and WT were filtered by FDR <0.05 and FC >1.5, and developmental DEGs between WT E14.5 and WT E18.5 were filtered by FDR <0.01 and FC >2. Genes in DEG heatmaps were ordered by fold changes between ENL[T] mutant and WT from high to low, or clustered to six groups by direction of DEGs. Z-score normalization was performed on each row of heatmaps by R (v3.6.1). *X-y* plots were generated by GraphPad Prism 10 with $\log_2 FC$ (fold change) as the *x*-axis and $-\log_{10} FDR$ as the *y*-axis. Pearson correlation coefficient (PCC) between Six2-ENL[T1] and Six2-ENL[T3] was calculated by R (v3.6.1) and plotted by GraphPad Prism 10. Gene Ontology (GO) biological process terms and KEGG pathways enrichment were done by DAVID 2021[89,90] and ranked by $-\log_{10} FDR$ values. Gene set enrichment analysis was done by GSEA (v4.3.2)[54]. Gene sets used for GSEA include MSigDB (https://www.gsea-msigdb.org/gsea/msigdb/), DEGs from HEK293 cells stably expressing ENL T1 or T3 mutants (derived from GSE1251866 followed the same analysis processes)[43], and DEGs identified in human Wilms tumors with vs. without *ENL* mutations in the TARGET dataset (RNA-seq data downloaded from TARGET-WT project, https://portal.gdc.cancer.gov/projects/TARGET-WT) (Supplementary Data 2, 19). Heatmaps of selected genes and pathways were generated by GraphPad Prism 10.

## Construction and sequencing of spatial gene expression libraries

Visium HD Spatial Gene Expression platform (10x Genomics) was used for whole-transcriptome spatial analysis at single cell-scale resolution. Visium HD slide processing, library generation, and sequencing were performed by the Van Andel Institute Histology and Genomics Cores. FFPE tissue sections were placed on slides, stained, and imaged according to the manufacturer's instructions in the Visium HD FFPE Tissue Preparation Handbook (10x Genomics, Rev A). Slides were further processed with the 10X Visium Mouse Transcriptome Probe Kit v2 (10x Genomics, Rev A) according to the manufacturer's instruction,

including probe transfer to the Visium HD slide (6.5 mm) using the Visium CytAssist (10x Genomics, Pleasanton, CA). The quality and quantity of the completed library pools were assessed using a combination of Agilent DNA High Sensitivity chip (Agilent Technologies, Inc.) and QuantiFluor® dsDNA System (Promega Corp.). 43 ×50, paired-end sequencing was performed on an Element AVITI sequencer using a 150 bp sequencing kit (Element Biosciences, San Diego, CA, USA) to an average depth of 480 M reads per capture area. Base calling was performed on the instrument and AVITI OS (v2.6.2) output was demultiplexed and converted to FastQ format with Element Biosciences Bases2fastq (v1.7.0).

### Data processing, filtering, and gene annotation

The SIX2 and FOXD1 datasets were analyzed separately for all steps of spatial transcriptomics analysis. Fastq files were processed using the SpaceRanger count pipeline in SpaceRanger v3.0.1 with the "refdata-gex-mm10-2020-A" mm10 index files provided by 10x Genomics and a JSON file generated using Loupe Browser v8.0.0 containing information for alignment of microscope images to the Visium HD slide. A custom probe designed to target the mutant allele of *Mllt1* was unsuccessful in distinguishing wildtype and mutant *Mllt1* alleles. This custom probe (TGAGCTTCTCACAGCGCAGGTGGTTCAGGTGGTTGA-CAGGAGGGTTGCCC) was appended to the "Visium_Mouse_Transcriptome_Probe_Set_v2.0_mm10-2020-A.csv" file. The custom probe was assigned to the same gene ID and name as the native probes so that the counts from the custom probe were counted together with those from the native *Mllt1* probes.

Downstream analyses were performed on the 8-μm binned output from SpaceRanger, which includes UMI gene counts per bin and physical location information for each bin relative to microscope images. To distinguish bins derived from different samples processed on the same slide capture area, bins were overlaid on the H&E microscope image for manual annotation based on sample tissue boundaries using Loupe Browser v8.0.0. At the same step, bins corresponding to debris were flagged for removal. Annotations made in the Loupe Browser were exported to comma-separated values (CSV) files, which were imported into R v4.4.0 along with the gene counts. Quality control, batch correction, dimension reduction, and clustering were performed using Seurat v5.1.0[91]. Bins annotated as debris or with <100 features detected or <200 UMI counts were removed. In the SIX2 dataset, 231,240 bins from eight samples were retained, with the per-sample bin count ranging from 25,610 to 31,364 bins. In the FOXD1 dataset, based on the spatial distribution of count numbers and preliminary clustering pattern, further analysis was focused on four samples (WT-1, WT-2, T1-1, and T1-2). From these four samples, 85,467 bins were retained, with the per-sample bin count ranging from 19,012 to 23,787 bins. UMI counts were normalized for total expression in each bin and log-transformed. Bins from different samples were split into separate 'layers', which informs the Seurat software which sample each bin belongs to. The top 2000 genes with the highest variation were identified for each sample with the "FindVariableFeatures" function and used to subset 5000 representative bins from each sample using the "SketchData" function, which implements a leverage score approach designed to retain rare cell populations[91]. For each "sketched" sample, the 2000 most variable genes were re-identified, which were scaled using the "ScaleData" function and used for a combined principal component analysis (PCA) of all samples using the "RunPCA" function. The first 50 principal components were used to perform uniform manifold approximation and projection (UMAP) for visualization using the "RunUMAP" function. The lack of overlap between samples in sections of the UMAP plot indicated a need for batch correction in both SIX2 and FOXD1 datasets.

### Batch correction and cell type annotation

A batch correction was performed on the PCA components for each "sketched" dataset using the "IntegrateLayers" function with the reciprocal PCA method[92]. The first 30 dimensions of the batch-corrected data were used for UMAP to visualize the results of the batch correction. A shared nearest neighbor (SNN) graph of the bins was constructed using the first 30 dimensions of the batch-corrected data and the "FindNeighbors" function; the SNN graph was used for unsupervised clustering of the bins using the "FindClusters" function with the Louvain algorithm and resolution set to 2. The batch correction was extended from the "sketched" bins to the entire dataset using the "ProjectIntegration" function. Similarly, the clusters identified in the "sketched" cells were projected onto the full dataset using the 'ProjectData' function with the parameter, "dims = 1:30". UMAP was performed on the first 30 dimensions of the batch-corrected full dataset using the "RunUMAP" function. Marker genes for each cluster were identified using the "FindConservedMarkers" function with the parameter, "min.cells.group = 10", which performs Wilcoxon rank-sum tests to compare bins in a given cluster to all bins not in that cluster in each sample separately, and combines the $P$ values for a given gene using the "minimump" method implemented in metap v1.11. Clusters were annotated to cell types and tissue substructures using the cluster marker genes, and by overlaying the bin clusters on the H&E microscope images; clusters annotated to the same label were merged. Using the recoded cluster labels, the marker gene analysis was repeated as described above.

### DEG identification in spatial transcriptomics data

Differential expression between WT and T1 for each cluster was performed using "seudobulked" counts and DESeq2 v1.45.3[93]. Specifically, the "AggregateExpression" was run to sum gene counts for all bins in each cluster-sample combination. Cluster-sample combinations derived from less than ten bins were removed, and differential expression analysis was run for the clusters with at least two replicate samples for each genotype. For differential expression analysis of each cluster, only genes with at least 10 counts in at least n samples were kept, where n is the lesser of the numbers of pseudobulked samples for T1 and WT. DESeq2, as implemented in the "FindMarkers" function, was run on these pseudobulked samples, and a significance cutoff of 0.05 adjusted $P$ value was used.

### Cell-cell communication analysis using LIANA

Ligand-receptor (L-R) cell-cell communication analyses were performed using LIANA+ v1.4.0[69,70]. Specifically, bin counts and bin physical locations from SpaceRanger were imported into Python v3.10.0 as an AnnData object[94]. Bin annotations from the R analysis above were merged to this object, allowing the object to be subsetted to specific samples. LIANA+ analyses were run on each sample separately to avoid batch effects. Using the pp.normalize_total and pp.log1p functions in ScanPy v1.10.4[95], counts were normalized to a sum of 10,000 and log-transformed. To identify co-expressed L-R pairs in the "mouse-consensus" database, compiled by the LIANA+ authors from multiple L-R databases, the "mt.rank_aggregate" function was run with the parameters, "resource_name = "mouseconsensus", expr_prop=0.05", grouping by the recoded bin clusters identified in the R analysis above. This calculates L-R scores for each L-R pair between each pair of clusters using multiple methods, namely CellPhoneDB, Connectome, NATMI, $\log_2$FC, and SingleCellSignalR[70,96–99]. The scores from the different tools were aggregated into a $P$ value for magnitude (strength of the interaction) and a $P$ value for specificity (compared to other pairs of clusters) of a given L-R interaction between two clusters using robust rank aggregation[100].

## Software used for spatial transcriptomics figures

Summary figures of the Seurat analysis were created in R. Spatial gene expression plots were generated using Loupe Browser v8.0.0. Co-expression spatial plots were plotted using the 'SpatialFeaturePlotBlend' wrapper (https://github.com/george-hall-ucl/SpatialFeaturePlotBlend; commit 342949 d) for "SpatialFeaturePlot" in Seurat. Violin plots were created using dittoSeq v1.17.0[101]. Heatmaps were created using ComplexHeatmap v2.21.1[102].

## Statistics and reproducibility

No statistical methods were used to predetermine the sample size. Experimental data were presented as mean ± s.d. unless stated otherwise. Statistical significance was calculated by two-tailed, unpaired Student's $t$-test on two experimental conditions with $P < 0.05$ considered statistically significant unless stated otherwise. A two-tailed Chi-square test was used for the quantification of cell type proportions. The Wald test was used in violin plots, comparing spatial gene expression in cell types. Robust rank aggregation was employed to calculate $P$ values for the magnitude and specificity of L-R interactions. A one-sided Fisher's exact test was used for enrichment analysis of GO biological process terms and KEGG pathways. The cumulative distribution function of the t-distribution was used for $P$ values in the Pearson correlation coefficient. The number of replicates and the statistical tests used are included in the corresponding figure legends.

## Reporting summary

Further information on research design is available in the Nature Portfolio Reporting Summary linked to this article.

## Data availability

Bulk RNA-seq data and spatial transcriptomic data have been deposited in the Gene Expression Omnibus database under accession numbers GSE266256 and GSE283433, respectively. All other raw data generated or analyzed during this study are included in this published article and its Supplementary Information files. Source data are provided with this paper.

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

## Acknowledgements

We thank members of the Wen laboratory, Shi laboratory, and Drs. Tao Yang and Bart Williams for scientific discussion throughout the study. The authors gratefully acknowledge the support and services provided by the Core Technologies and Service at Van Andel Institute, including the Genomics Core (RRID:SCR_022913) for NGS library construction and sequencing, the Flow Cytometry Core (RRID:SCR_022685) for cell sorting, the Vivarium (RRID:SCR_023211) for maintenance of mouse colonies, the Pathology and Biorepository Core (RRID:SCR_022912) for assistance with tissue sample processing and staining. We thank Galen Hostetter, the pathologist at the pathology core, for histological evaluation. Computation for the work described in this manuscript was supported by the High-Performance Cluster and Cloud Computing (HPC3) resource at the VAI. This work was supported in part by funds from Van Andel Institute (to H.W.), NIH/NCI awards R01CA260666 and R01CA255506 (to H.W.), and R01CA268440 (to X.S.). H.W. is a Scholar of The Leukemia & Lymphoma Society.

## Author contributions

Z.X. and H.X. contributed equally to this work. H.W. and X.S. conceived the project. H.W. and Z.X. designed experiments in this study. Z.X., Y.S., and K. Li performed mouse work; Z.X. sorted cells and performed RNA-seq experiments; Z.X., M.W., L.T., and M.A. performed spatial transcriptomic study; H.X. performed all bulk genomic data analyses; K. Lau performed spatial transcriptomic data analysis. H.W., X.S., Z.X., and H.X. wrote the paper with critical inputs from all authors. H.W. supervised the overall study.

## Competing interests

The authors declare no competing interests.
