## [Peer Review file · Nature Communications]

Expression of ENL YEATS domain tumor mutations in nephrogenic or stromal lineage impairs kidney development

Corresponding Author: Dr Hong Wen

Version 0:

Reviewer comments:

Reviewer #1

(Remarks to the Author)

Xue et al., presented a study on trying to model ENL mutants in Wilms tumor model. It is established that ENL mutation represents gain-of-function mutations on controlling gene expression in Wilms tumor but the lack of faithful models pose an important challenge in the field. This elegant study took exhaustive approaches and great efforts on generating various mouse models mimicking patient derived ENL mutations to address this question. This study uses the whole-body mutant, Six2+ nephrogenic mutant as well as Foxd1+ stromal lineage mutant and showed that ENL mutations yielded consistent kidney development defect and lethality. In addition, this study also uses extensive transcriptomic analysis and showed that gene expression changes induced by ENL mutations recapitulate the patient population carrying such mutations. The data presented are all very convincing and high quality. In addition, although the model cannot be used to model Wilms tumor development at this stage, it produces very useful information for the field. This paper should be acceptable with very minor revision. The discussion of potential future modeling combining ENL and other mutations in Wilms tumor may be considered.

Reviewer #2

(Remarks to the Author)

Xue et al. describe the analysis of mice with ENL mutations. Mutations in the nephron lineage resulted in premature commitment of progenitors and differentiation block, accompanied by upregulation of Hox genes and Wnt-related genes, while mutations in the stromal lineage led to expansion of nephron progenitors, a typical stroma progenitor-less phenotype. These reported phenotypes are informative in developmental biology and tumor biology. However, the data presented are largely based on whole kidney bulk RNA-seq and the analysis ends at the stage where some candidate genes are picked up. The manuscript lacks subsequent histological validation of the affected genes and the precise identification of the responsible cell types remains unclear. In addition, the functional involvement of these candidate genes in relation to the phenotypes is not examined, leaving the mechanistic insights hypothetical. The analysis in the stroma mutant mice is more incomplete to provide a solid molecular identity of the "stroma-nephron crosstalk/interaction".

Major comments

1. Line 190-192, 202: There are many descriptions based on HE-stained sections, such as thinner nephron progenitor or reduced number of nascent nephrons (CSB/SSB), but they are not as reliable as the immunostaining data. In addition, most of the description in Figure 2 overlaps with that based on the immunostaining data in Figure 3. The authors should reduce the explanation of the HE sections and combine them with the immunostaining data. Quantitative data using HE sections in Figure 2 can also be replaced with quantification data using immunostained sections.
2. Line 233, 242: Although the authors claim that nephrogenesis is blocked at multiple stages, this conclusion is superficial. The HE-based method to detect nascent nephrons (CSB and SSB) described on line 689 is also not very reliable. Antibodies against Lhx1 and Jag1 can be used as markers for nascent nephrons to precisely identify the differentiation blocking points. scRNA-seq analysis will also be helpful to identify the altered identity of differentiating nephrons at a higher resolution.
3. Line 169, 245: In Six2 TGC mice, the transgene is not integrated into the Rosa26 locus. The BAC transgene is inserted into chromosome 1 and aberrant Six3 expression occurs (Pert et al. J Am Soc Nephrol 35: 566-577, 2024), explaining the fact that nephron progenitor reduction occurs even in the presence of one copy of Six2-TGCTg and the kidney becomes dramatically small in the presence of two copies. Therefore, nephron progenitors could be more accurately compared

between Six2-TGCtg/+; ENL+/+ and Six2-TGCtg/+; ENL T/+, rather than between ENL+/+ and Six2-TGCtg/+; ENL T/+. At least, the authors should mention this point in the text.

4. RNA-seq analysis of whole kidney (Fig. 4) and sorted nephron cells (Fig. 5) have a lot of overlap and lead to a similar conclusion. These two figures can be combined or Figure 4 can be moved to a supplementary figure. In either case, the authors should reduce the description of the main text before reaching the conclusion: upregulation of Hox genes and Wnt-related genes. In addition, the upregulation of these genes should be validated histologically by immunostaining or in situ hybridization to confirm the correctness of the computational analysis and to identify the cell types that express the key genes.

5. Line 344: The upregulation of Hox genes is interesting, but in a way it was anticipated by the other studies. Is there any evidence that Hox upregulation is the cause of the nephron defects in the mutant mice? I also wonder if Wnt signaling is really increased. The authors could examine the expression of Wnt signaling indicators such as Axin2 and Lef1 by RNA-seq and immunostaining. Does Wnt signaling really explain the nephron defects in the mutant mice? Which is more important, Hox genes or Wnt-related genes? In other words, the functional relevance of the candidate genes should be considered more carefully and solid mechanistic insights should be presented.

6. Figures 6&7: Again, there is a lot of overlap in the conclusions based on HE-stained sections and immunostained sections. Reduce the description and add more detailed staining of stromal progenitor markers such as Foxd1 and Meis1/2 to examine the reduction of stromal progenitors.

7. The nephron expansion in the stromal lineage-specific mutant mice is a typical phenotype observed in mice lacking stromal progenitors. In the current manuscript, the molecular identity of the "stroma-nephron crosstalk/interaction" remains unresolved despite the RNA-seq analysis in Figure 8. However, some genes such as Fat4 and Dchs1 have been reported to be involved in this process. The authors should review the literature and narrow down the candidate genes/pathways as they did in the nephron lineage. Sorting of stromal cells or scRNA-seq may also be helpful, as bulk analysis of the entire kidney may miss the changes in stromal progenitors that exist in a small number of cells per kidney. Of course, the resulting candidate genes, including many of those mentioned on pages 18-19, should be validated histologically and functionally. In particular, it is important to determine which cells express which ligands/receptors to reveal the "stroma-nephron crosstalk/interaction".

8. As the authors admit, there is no tumor formation in the mutant mice, which remains the weak point of this manuscript.

Reviewer #3

(Remarks to the Author)

The Xue et al "Expression of ENL YEATS domain tumor mutations in nephrogenic or stromal lineage impairs kidney development" demonstrates that expression of either of two cancer-associated gain-of-function ENL mutations during development impair kidney development by different chromatin-mediated mechanisms. Specifically, the authors utilize ENL mutant GEMMs to investigate the consequence in development for two cancer-associated ENL mutations, finding that heterozygous expression leads to neonatal lethality due to distinct kidney development defects. Transcriptomic analyses identify specific gene expression pathways that are deregulated by the ENL mutants; notably, different pathways (e.g. Hox genes and Wnt signaling vs other developmental programs) are regulated by the different mutants, highlighting the complexity of chromatin regulation in governing transcriptional programs in development and how deregulation in these mechanisms can lead to oncogenesis.

Overall, this is an important study that advances our understanding of ENL driver mutations in pediatric cancer. These studies also provide a foundation for future work in which conditional expression of the various mutants post-development may provide novel tumor models to understand Wilms tumor.

Comments:

1. While I think it is beyond the scope of the present manuscript, it would be helpful if the authors comment on the technical limitations that prevent chromatin binding studies of ENL in their mouse models.

Version 1:

Reviewer comments:

Reviewer #2

(Remarks to the Author)

The authors have substantially revised the manuscript and addressed all my concerns. The authors added the spatial transcriptome analyses of the mutant mice, followed by histological validation, and provided the molecular candidates for stroma-nephron crosstalk/interaction. In particular, the spatial transcriptome data are of high quality and will have an impact on the research community.

Some minor comments to be corrected

1. Page 7, line 159: not collecting tubules but collecting ducts
2. Page 26, line 706: please cite the reference.

Responses to Reviewers' Comments

We sincerely thank all reviewers for their insightful comments and suggestions. Based on the collective suggestions, we have now revised the manuscript and incorporated several sets of new data to further strengthen our findings. We have performed RNA-seq analysis on sorted ENL^{T1} mutant stromal cells to identify transcriptional changes specific to the mutant stromal lineage. Furthermore, we have conducted spatial transcriptomic analysis in both Six2-ENL^{T1} and Foxd1-ENL^{T1} kidneys. This approach enables whole-transcriptome spatial analysis at single-cell resolution, allowing us to identify transcriptomic changes caused by ENL^T mutant in specific cell types and renal structures. Additionally, we have expanded our immunostaining analyses of multiple progenitor and nephron structure markers, providing more precise phenotype quantification and validation of the transcriptional changes at the protein level. Below, please find our point-by-point responses to all comments.

Reviewer #1 (Remarks to the Author):

Xue et al., presented a study on trying to model ENL mutants in Wilms tumor model. It is established that ENL mutation represents gain-of-function mutations on controlling gene expression in Wilms tumor but the lack of faithful models pose an important challenge in the field. This elegant study took exhaustive approaches and great efforts on generating various mouse models mimicking patient derived ENL mutations to address this question. This study uses the whole-body mutant, Six2+ nephrogenic mutant as well as Foxd1+ stromal lineage mutant and showed that ENL mutations yielded consistent kidney development defect and lethality. In addition, this study also uses extensive transcriptomic analysis and showed that gene expression changes induced by ENL mutations recapitulate the patient population carrying such mutations. The data presented are all very convincing and high quality. In addition, although the model cannot be used to model Wilms tumor development at this stage, it produces very useful information for the field. This paper should be acceptable with very minor revision. The discussion of potential future modeling combining ENL and other mutations in Wilms tumor may be considered.

Thank you for finding our work interesting and high quality and thanks for the great suggestion. We have added a discussion of potential future modelling combining ENL YEATS mutations with other concurrent but rare alterations found in human Wilms' tumor cases (lines 693-695). These include *CTNNB1* mutations and biallelic *Igf2* expression, resulting from loss of imprinting (LOI) at ICR1 on 11p15^{1,2}.

Reviewer #2 (Remarks to the Author):

Xue et al. describe the analysis of mice with ENL mutations. Mutations in the nephron lineage resulted in premature commitment of progenitors and differentiation block, accompanied by upregulation of Hox genes and Wnt-related genes, while mutations in the stromal lineage led to expansion of nephron progenitors, a typical stroma progenitor-less phenotype. These reported phenotypes are informative in developmental biology and tumor biology. However, the data presented are largely based on whole kidney bulk RNA-seq and the analysis ends at the stage where some candidate genes are picked up. The manuscript lacks subsequent histological validation of the affected genes and the precise identification of the responsible cell types remains unclear. In addition, the functional involvement of these candidate genes in relation to the phenotypes is not examined, leaving the mechanistic insights hypothetical. The analysis in

the stroma mutant mice is more incomplete to provide a solid molecular identity of the "stroma-nephron crosstalk/interaction".

Major comments

1. Line 190-192, 202: There are many descriptions based on HE-stained sections, such as thinner nephron progenitor or reduced number of nascent nephrons (CSB/SSB), but they are not as reliable as the immunostaining data. In addition, most of the description in Figure 2 overlaps with that based on the immunostaining data in Figure 3. The authors should reduce the explanation of the HE sections and combine them with the immunostaining data. Quantitative data using HE sections in Figure 2 can also be replaced with quantification data using immunostained sections.

Thank you for the great suggestions. As suggested, we have combined Figures 2 and 3 into the new Figure 2 and replaced H&E quantification data with quantification data derived from immunostaining sections. Specifically, we used SIX2 and KRT8 to label the cap mesenchyme and ureteric bud, LTL/E-Cad for the proximal tubule, SLC12A3/E-Cad for the distal tubule, and WT1/PODXL for the glomeruli. RV and nascent nephrons (CSB and SSB) were stained with NCAM and WT1 for quantification. Immunofluorescence staining images and nephron structure quantifications for the Six2-ENL^{T1} mutant are now included in new Figure 2f-v and Figure S6, while data for the Six2-ENL^{T3} mutant are presented in new Figures S7 and S8.

2. Line 233, 242: Although the authors claim that nephrogenesis is blocked at multiple stages, this conclusion is superficial. The HE-based method to detect nascent nephrons (CSB and SSB) described on line 689 is also not very reliable. Antibodies against Lhx1 and Jag1 can be used as markers for nascent nephrons to precisely identify the differentiation blocking points. scRNA-seq analysis will also be helpful to identify the altered identity of differentiating nephrons at a higher resolution.

Thank you for the suggestions. We performed immunostaining for WT1 and NCAM to detect RV and nascent nephrons (Figure 2h-i, Figure S6g-l, and Figure S7e-h). Compared to ENL^{WT} controls, Six2-ENL^T mutant embryonic kidneys exhibited lower WT1 and NCAM signals in the cap mesenchyme and a reduced number of WT1⁺ NCAM⁺ RV and nascent nephron structures. In addition, although early podocytes in glomerular cleft were observed in Six2-ENL^T mutants, they failed to further develop into more mature glomeruli beyond this stage. These phenotypes suggest that expression of ENL^T mutants in the nephron lineage impedes nephrogenesis at multiple stages (**Figure R1**).

Figure R1 (Figure S6g-h). Nephrogenesis defects in Six2-ENL^{T1} kidneys. g, h, Immunostaining of WT1 and NCAM to detect nascent nephron in E18.5 ENL^{WT} and Six2-ENL^{T1} kidneys. CM, cap mesenchyme; RV, renal vesical; S, S-shaped body; G, glomerulus. Dash lines indicate CM in Six2-ENL^{T1} mutant. Dash line circles indicate cystic glomeruli. Arrows indicate abnormal differentiating nephrons in Six2-ENL^{T1} mutant kidney. Scale bars, 100 μm.

Moreover, we conducted spatial transcriptomic analysis on Six2-ENL^{T1} and control kidney sections using the Visium HD Spatial Gene Expression platform. This approach not only profiles the transcriptome at single cell-scale resolution but also defines gene expression changes in specific cell types and anatomic renal structures within their tissue context. In Six2-ENL^{T1} mutant kidneys, we observed:

- 1) Aberrant activation of numerous *Hox* genes across all nephron cell types (Figure 4d-f and Figure S12c), suggesting extensive and prolonged activation (**Figure R2**).
- 2) Early activation of NP commitment genes (e.g., *Wnt4* and *Pax8*), accompanied by downregulation of NP “stemness” genes (e.g., *Cited1* and *Six2*) in the cap mesenchyme (Figure 4e, f; and Figure S12d, f, g), potentially leading to reduced NP self-renewal and premature NP commitment (**Figure R3**).
- 3) Certain NP commitment genes (e.g., *Tmem100* and *Snap91*) were not fully activated in committing NPs (PTA and RV), while others were expressed at higher levels than in controls in nascent nephrons (Figure S12d, f, g), potentially disrupting proper differentiation.
- 4) Downregulation of marker genes for different nascent nephron segments (e.g. *Lhx1*, *Mafb*, and *Hey1*) (Figure 4d and Figure S12d, f), indicating further differentiation delay or blockade.

Together, the spatial transcriptomic analysis provides the molecular basis for impeded nephrogenesis at multiple stages in Six2-ENL^T mutants. The new data are now included in the new Figure 4; Figures S11 and S12; and Supplementary Tables 9-11.

Additionally, we validated the reduced protein levels of LHX1 in nascent nephrons of Six2-ENL^{T1} mutant by immunostaining (Figure S12i and **Figure R4**).

3. Line 169, 245: In Six2 TGC mice, the transgene is not integrated into the Rosa26 locus. The BAC transgene is inserted into chromosome 1 and aberrant Six3 expression occurs (Pert et al. J Am Soc Nephrol 35: 566-577, 2024), explaining the fact that nephron progenitor reduction occurs even in the presence of one copy of Six2-TGCTg and the kidney becomes dramatically small in the presence of two copies. Therefore, nephron progenitors could be more accurately compared between Six2-TGCTg/+; ENL+/+ and Six2-TGCTg/+; ENL T/+, rather than between ENL+/+ and Six2-TGCTg/+; ENL T/+. At least, the authors should mention this point in the text.

Thank you for the information and great suggestions. We have corrected the information regarding the integration site of the transgene (lines 174-176) and included a discussion on the aberrant Six3 expression and this Cre strain (lines 187-192). In our breeding setup using Six2-TGC^{tg/+} and Enl^{T-fl/+} mice to generate progeny, we observed obviously smaller kidney at P0 only in Six2-TGC^{tg/+}; Enl^{T-fl/+} mutants, but not in Six2-TGC^{tg/+}; Enl^{+/+} controls (Figure 2b-e and Figure S4a-e). As reported in the referred reference³, we have confirmed the ectopic Six3 expression in Six2-TGC^{tg/+}; Enl^{+/+} and Six2-TGC^{tg/+}; Enl^{T-fl/+} kidneys by RT-PCR. We have also quantified cap mesenchyme and CSB/SSB in E18.5 embryonic kidneys from progeny of Six2-TGC^{tg/+} and Enl^{T-fl/+} mating. No significant differences were observed between Six2-TGC^{tg/+} and wildtype kidneys (Figure R5). These new data are now included in Figure S4h-i. As suggested, we used Six2-TGC^{tg/+}; Enl^{+/+} kidneys as controls in our study.

4. RNA-seq analysis of whole kidney (Fig. 4) and sorted nephron cells (Fig. 5) have a lot of overlap and lead to a similar conclusion. These two figures can be combined or Figure 4 can be moved to a supplementary figure. In either case, the authors should reduce the description of the main text before reaching the conclusion: upregulation of Hox genes and Wnt-related genes.

In addition, the upregulation of these genes should be validated histologically by immunostaining or in situ hybridization to confirm the correctness of the computational analysis and to identify the cell types that express the key genes.

Thank you for the suggestions. We have now moved Figure 4 to the Supplementary Information as Figure S9 and reduced description of this part in the main text. Aberrant activation of *Hox* genes was validated in our new spatial transcriptomic analysis across all nephron cell types in mutant kidneys (Figure 4d-f and Figure S12c). In addition, genes upregulated in the mutant NP population were enriched in the Wnt signaling pathway (Figure 4c and Figure S12e). Notably, early activation of *Wnt4* was detected in NPs and was further upregulated in committing NPs of Six2-ENL^{T1} mutant kidneys (Figure 4d-f). *Lef1*, a Wnt target gene, was also upregulated in NPs and committing NPs (Figure S12e). As suggested, we have confirmed elevated LEF1 protein levels in mutant NP, PTA and RV by immunostaining (Figure S12h, as shown in **Figure R6**).

5. Line 344: The upregulation of Hox genes is interesting, but in a way it was anticipated by the other studies. Is there any evidence that Hox upregulation is the cause of the nephron defects in the mutant mice? I also wonder if Wnt signaling is really increased. The authors could examine the expression of Wnt signaling indicators such as Axin2 and Lef1 by RNA-seq and immunostaining. Does Wnt signaling really explain the nephron defects in the mutant mice? Which is more important, Hox genes or Wnt-related genes? In other words, the functional relevance of the candidate genes should be considered more carefully and solid mechanistic insights should be presented.

These are great points. Beyond *Hox* gene activation, upregulation of Wnt signaling is indeed a signature of ENL^T mutant nephron progenitors. This conclusion is supported by evidence from bulk RNA-seq, spatial transcriptomic analysis, and immunostaining of LEF1, a key indicator of Wnt signaling (Figures 3, 4; and Figures S9, S10, and S12). Notably, *Wnt4* is not the only NP commitment gene aberrantly activated in mutant NPs—others upregulated genes include *Pax8*, *Rdh10*, and *Clu*.

Our previous study on ENL^T mutations in human cell lines identified *HOX* genes as direct targets of both wildtype and mutant ENL, with ENL^T mutations driving aberrant *HOX* gene activation⁴. In that study, ENL^{T1} binding was not detected at the *WNT4* locus, suggesting that ENL or ENL mutants may not directly regulate *WNT4*. Interestingly, it has been reported that *Wnt4* is a direct target of HOXD11 and its paralogs in mouse kidney⁵. In our pilot experiments using HEK293 cells, overexpression of several *HOX* genes led to mild activation of Wnt-related

genes (**Figure R7**). Based on these findings, we speculate that ENL^T mutants directly activate *Hox* genes, which in turn drive *Wnt4* expression. However, due to the lack of an antibody recognizing mouse ENL at endogenous levels, it remains challenging to distinguish between direct and secondary effects of ENL mutations in our mouse models. We have acknowledged this technical limitation in the main text (lines 640-642). The development of new reagents will be necessary for future studies to fully elucidate the hierarchy of gene expression changes driven by ENL mutations.

Hox genes, particularly the posterior *Hox* genes, play a crucial role in early kidney development⁶⁻⁸. Likewise, Wnt signaling is also essential for kidney development by regulating various aspects of nephrogenesis, including the initial induction of the metanephric mesenchyme, ureteric bud branching, renal vesicle induction, and nephron maturation⁹⁻¹¹. Given the important roles of these genes in kidney development, we hypothesize that the activation of *Hox* genes, Wnt-related genes and other NP commitment genes, along with the downregulation of essential nephrogenesis genes across NP and different nascent nephron segments, collectively contribute to the developmental defects observed in ENL^T mutant kidneys. In this context, dissecting the functional relevance of individual candidate genes presents a significant challenge and would require substantial efforts in future studies.

6. Figures 6&7: Again, there is a lot of overlap in the conclusions based on HE-stained sections and immunostained sections. Reduce the description and add more detailed staining of stromal progenitor markers such as *Foxd1* and *Meis1/2* to examine the reduction of stromal progenitors.

Thank you. We have now combined Figures 6 and 7 as new Figure 5 and reduced the description. Nephron quantification data have been replaced with new results based on immunostaining (Figure 5d-v for *Foxd1*-ENL^{T1} and Figure S14d-v for *Foxd1*-ENL^{T3}).

Due to the lack of reliable antibody recognizing FOXD1, we have instead used immunostaining of stromal marker TENASCIN to quantify the thickness of capsular stroma (Figure 5h-i, Figure S13i-j, and Figure S14h-i). As suggested, we have also added immunostaining of another stromal marker MEIS1/2 for validation (Figure S13p). These new results demonstrate a thickened capsular stroma in *Foxd1*-ENL^T mutant kidneys (**Figure R8**).

7. The nephron expansion in the stromal lineage-specific mutant mice is a typical phenotype observed in mice lacking stromal progenitors. In the current manuscript, the molecular identity of the "stroma-nephron crosstalk/interaction" remains unresolved despite the RNA-seq analysis in Figure 8. However, some genes such as *Fat4* and *Dchs1* have been reported to be involved in this process. The authors should review the literature and narrow down the candidate genes/pathways as they did in the nephron lineage. Sorting of stromal cells or scRNA-seq may also be helpful, as bulk analysis of the entire kidney may miss the changes in stromal progenitors that exist in a small number of cells per kidney. Of course, the resulting candidate genes, including many of those mentioned on pages 18-19, should be validated histologically and functionally. In particular, it is important to determine which cells express which ligands/receptors to reveal the "stroma-nephron crosstalk/interaction".

This is an important point and thank you for the suggestions. We have performed RNA-seq analysis on sorted ENL^{T1} mutant stromal cells and replaced the previous whole-kidney RNA-seq data (new Figure 6). This allowed us to specifically determine gene expression changes caused by ENL^{T1} mutant expression in the stromal lineage. This new analysis revealed that genes upregulated in mutant stromal cells were enriched in developmental and cell adhesion-associated processes. Similar to ENL^{T1} mutant cells in the nephron lineage, *Hox* gene activation is a prominent feature in mutant stromal cells. While a few stromal marker genes—such as *Vcam1*, *Nts*, *Meis2*, and *Fn1*—were upregulated, more stromal genes were downregulated, including *Pbx1*, *Tcf21*, *Pdfrb*, *Ecm1*, and *Agtr2*. These genes are known to be associated with aberrant nephrogenesis and ureteric branching when perturbed. These new data are now included in new Figure 6 and Figure S15.

Moreover, we have conducted spatial transcriptomic analysis on Foxd1-ENL^{T1} mutant and control kidney sections to examine ligand and receptor gene expression changes in specific cell types and renal structures. Spatial transcriptome profiling revealed that ENL^{T1} expression in the stromal lineage primarily affected local gene expression, and expansion of the NP and committing NP population was confirmed. While nephron expansion is a typical phenotype in mice lacking stromal progenitors, we did not detect expression changes in *Fat4* or *Dchs1* in either bulk RNA-seq or spatial transcriptomic analysis. This has been discussed in the main text (Lines 662-666). Instead, we identified several ligand-encoding genes—including *Fn1*, *Thbs1*, *Col4a2*, *Gpc3* and *Ntn1*—that were either upregulated or downregulated in ENL^{T1} mutant stroma (Figure S18a, b). Previous studies indicate that these ligands are important for kidney development, and their dysregulation leads to various developmental defects. For instance, deletion of *Fn1* in cultured metanephric mouse kidneys reduces UB epithelial branching and

kidney size¹². We observed the expression of *Fn1* and its engaged receptor *Itga8* localized in adjacent domains within the nephrogenic zone (NGZ) stroma and NP, respectively (Figure 7e). Increased FN protein level and its localization adjacent to ITGA8 in cap mesenchyme were confirmed by immunostaining (Figure 7f). Consistent with the up- and down-regulation of these ligands, corresponding ligand-receptor (L-R) interactions were either increased or decreased in the ENL^{T1} mutant (Figure 7g and Figure S18f-k). In addition, our new results suggest that altered L-R interactions potentially affect both stroma-NP and stroma-UB interactions, indicating that stromal ENL^{T1} mutation may disrupt nephrogenesis involving both NP and UB compartments. These new data are now included in new Figures 7; Figures S16-18; and Supplementary Tables 12-16, and as shown in **Figure R9** below. It would be interesting to investigate in the future how dysregulated stroma-epithelium crosstalk through these L-R pairs contributes to nephrogenesis defects.

8. As the authors admit, there is no tumor formation in the mutant mice, which remains the weak point of this manuscript.

We discussed the limitations of our study and potential strategies to promote tumor formation in our mouse models in future research (lines 684-692). Furthermore, we have also discussed the possibilities of combining ENL mutations with other concurrent but rare alterations found in human Wilms' tumor cases for future studies (lines 693-695).

Reviewer #3 (Remarks to the Author):

The Xue et al "Expression of ENL YEATS domain tumor mutations in nephrogenic or stromal lineage impairs kidney development" demonstrates that expression of either of two cancer-associated gain-of-function ENL mutations during development impair kidney development by different chromatin-mediated mechanisms. Specifically, the authors utilize ENL mutant GEMMs to investigate the consequence in development for two cancer-associated ENL mutations, finding that heterozygous expression leads to neonatal lethality due to distinct kidney development defects. Transcriptomic analyses identify specific gene expression pathways that are deregulated by the ENL mutants; notably, different pathways (e.g. Hox genes and Wnt signaling vs other developmental programs) are regulated by the different mutants, highlighting the complexity of chromatin regulation in governing transcriptional programs in development

and how deregulation in these mechanisms can lead to oncogenesis.

Overall, this is an important study that advances our understanding of ENL driver mutations in pediatric cancer. These studies also provide a foundation for future work in which conditional expression of the various mutants post-development may provide novel tumor models to understand Wilms tumor.

Comments:

1. While I think it is beyond the scope of the present manuscript, it would be helpful if the authors comment on the technical limitations that prevent chromatin binding studies of ENL in their mouse models.

Thank you for finding our work important and thanks for the great suggestion. To investigate the molecular mechanism of mutant ENL in regulating gene expression in mouse kidneys, we screened several commercially available ENL antibodies to identify reagents suitable for profiling ENL chromatin binding by ChIP or CUT&RUN experiments. Unfortunately, all tested antibodies exhibited low sensitivity to mouse ENL and failed to detect mouse ENL at endogenous levels by Western blot or immunohistochemistry. We have added a discussion of this technical limitation in the mouse models (lines 640-642).

References:

- 1 Gadd, S. *et al.* A Children's Oncology Group and TARGET initiative exploring the genetic landscape of Wilms tumor. *Nat Genet* **49**, 1487-1494 (2017). <https://doi.org:10.1038/ng.3940>
- 2 Perlman, E. J. *et al.* MLLT1 YEATS domain mutations in clinically distinctive Favourable Histology Wilms tumours. *Nat Commun* **6**, 10013 (2015). <https://doi.org:10.1038/ncomms10013>
- 3 Perl, A. J. *et al.* Reduced Nephron Endowment in Six2-TGCTg Mice Is Due to Six3 Misexpression by Aberrant Enhancer-Promoter Interactions in the Transgene. *J Am Soc Nephrol* **35**, 566-577 (2024). <https://doi.org:10.1681/ASN.0000000000000324>
- 4 Wan, L. *et al.* Impaired cell fate through gain-of-function mutations in a chromatin reader. *Nature* **577**, 121-126 (2020). <https://doi.org:10.1038/s41586-019-1842-7>
- 5 O'Brien, L. L. *et al.* Transcriptional regulatory control of mammalian nephron progenitors revealed by multi-factor cistromic analysis and genetic studies. *PLoS Genet* **14**, e1007181 (2018). <https://doi.org:10.1371/journal.pgen.1007181>
- 6 Drake, K. A., Adam, M., Mahoney, R. & Potter, S. S. Disruption of Hox9,10,11 function results in cellular level lineage infidelity in the kidney. *Sci Rep* **8**, 6306 (2018). <https://doi.org:10.1038/s41598-018-24782-5>
- 7 Patterson, L. T., Pembaur, M. & Potter, S. S. Hoxa11 and Hoxd11 regulate branching morphogenesis of the ureteric bud in the developing kidney. *Development* **128**, 2153-2161 (2001). <https://doi.org:10.1242/dev.128.11.2153>
- 8 Yallowitz, A. R., Hrycaj, S. M., Short, K. M., Smyth, I. M. & Wellik, D. M. Hox10 genes function in kidney development in the differentiation and integration of the cortical stroma. *PLoS One* **6**, e23410 (2011). <https://doi.org:10.1371/journal.pone.0023410>
- 9 McMahon, A. P. Development of the Mammalian Kidney. *Curr Top Dev Biol* **117**, 31-64 (2016). <https://doi.org:10.1016/bs.ctdb.2015.10.010>
- 10 Meng, P., Zhu, M., Ling, X. & Zhou, L. Wnt signaling in kidney: the initiator or terminator? *J Mol Med (Berl)* **98**, 1511-1523 (2020). <https://doi.org:10.1007/s00109-020-01978-9>
- 11 Park, J. S., Valerius, M. T. & McMahon, A. P. Wnt/beta-catenin signaling regulates nephron induction during mouse kidney development. *Development* **134**, 2533-2539 (2007). <https://doi.org:10.1242/dev.006155>
- 12 Skoczynski, K. *et al.* The extracellular matrix protein fibronectin promotes metanephric kidney development. *Pflugers Arch* **476**, 963-974 (2024). <https://doi.org:10.1007/s00424-024-02954-9>

Responses to Reviewers' Comments

Reviewer #2 (Remarks to the Author):

The authors have substantially revised the manuscript and addressed all my concerns. The authors added the spatial transcriptome analyses of the mutant mice, followed by histological validation, and provided the molecular candidates for stroma-nephron crosstalk/interaction. In particular, the spatial transcriptome data are of high quality and will have an impact on the research community.

Some minor comments to be corrected

1. Page 7, line 159: not collecting tubules but collecting ducts
2. Page 26, line 706: please cite the reference.

Thank you. We have made the correction and cited the reference as suggested.